# Variational Causal Dynamics: Discovering Modular World Models from Interventions

## Abstract

Latent world models allow agents to reason about complex environments with high-dimensional observations. However, adapting to new environments and effectively leveraging previous knowledge remain significant challenges. We present *variational causal dynamics* (VCD), a structured world model that exploits the invariance of causal mechanisms across environments to achieve fast and modular adaptation. By causally factorising a transition model, VCD is able to identify reusable components across different environments. This is achieved by combining causal discovery and variational inference to learn a latent representation and transition model jointly in an unsupervised manner. Specifically, we optimise the evidence lower bound jointly over a representation model and a transition model structured as a causal graphical model. In evaluations on simulated environments with state and image observations, we show that VCD is able to successfully identify causal variables, and to discover consistent causal structures across different environments. Moreover, given a small number of observations in a previously unseen, intervened environment, VCD is able to identify the sparse changes in the dynamics and to adapt efficiently. In doing so, VCD significantly extends the capabilities of the current state-of-the-art in latent world models while also comparing favourably in terms of prediction accuracy.

## 1 Introduction

The ability to adapt flexibly and efficiently to novel environments is one of the most distinctive and compelling features of the human mind. It has been suggested that humans do so by learning internal models which not only contain abstract representations of the world, but also encode generalisable, structural relationships within the environment (Behrens et al., 2018). It is conjectured that this latter aspect is what allows humans to adapt efficiently and selectively. Recent efforts have been made to mimic this kind of representation in machine learning. *World models* (e.g. Ha and Schmidhuber, 2018) aim to capture the dynamics of an environment by distilling past experience into a parametric predictive model. Advances in latent variable models have enabled the learning of world models in a compact latent space (Ha and Schmidhuber, 2018; Watter et al., 2015; Hafner et al., 2019b; Buesing et al., 2018; Zhang et al., 2019) from high-dimensional observations such as images. Whilst these models have enabled agents to act in complex environments via planning (e.g. Hafner et al., 2019b; Sekar et al., 2020) or learning parametric policies (e.g. Hafner et al., 2019a; Ha and Schmidhuber, 2018), structurally adapting to changes in the environment remains a significant challenge. The consequence of this limitation is particularly pronounced when deploying learning agents to environments, where distribution shifts occur. As such, we argue that it is beneficial to build structural world models that afford modular and efficient adaptation, and that *causal* modeling offers a tantalising prospect to discover such structure from observations.

Causality plays a central role in understanding distribution changes, which can be modelled as causal interventions (Schölkopf et al., 2021). The Sparse Mechanism Shift hypothesis (Schölkopf et al., 2021; Bengio et al., 2019) (SMS) states that naturally occurring shifts in the data distribution can be attributed to sparse and local changes in the causal generative process. This implies that many causal mechanisms remain *invariant* across domains (Schölkopf et al., 2012; Peters et al., 2016; Zhang et al., 2015). In this light, learning a *causal* model of the environment enables agents to reason about distribution shifts and to exploit the invariance of learnt causal mechanisms across different environments. Hence, we posit that world models with a causal structure can facilitate modular

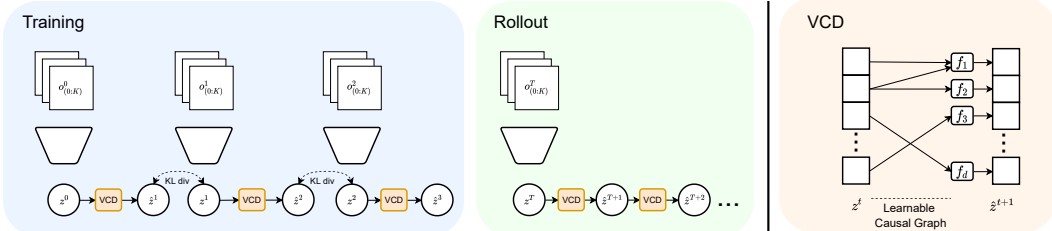

Figure 1: Left: The general architecture of VCD. The dynamics model is trained using observation sequences from multiple environments by minimising the KL divergence between the predicted state distribution and the encoded posterior state distribution. Rollouts in the latent space can be performed by recursively applying the learnt transition model. Right: The structure of the causal transition model. Each dimension of the latent space is treated as a causal variable. Predictions are made using only the causal parents of each variable, according to a *learnt* causal graph.

transfer of knowledge. To date, however, methods for causal discovery (Spirtes et al., 2000; Pearl, 2009; Peters et al., 2017; Brouillard et al., 2020; Ke et al., 2020) require access to abstract causal variables to learn causal models from data. These are not typically available in the context of world model learning, where we wish to operate directly on high-dimensional observations.

In order to benefit from the structure of causal models and the ability to represent high-dimensional observations, we propose *Variational Causal Dynamics* (VCD), which combines causal discovery with variational inference. Specifically, we train a latent state-space model with a structural transition model using variational inference and sparsity regularisation from causal discovery. By jointly training a representation and a transition model, VCD learns a causally factorised world model that can modularly adapt to different environments. The key intuition behind our approach is that, since sparse causal structures can only be discovered on abstract causal variables, training the representation and the causal discovery module in an end-to-end manner acts as an inductive bias that encourages causally meaningful representations. By leveraging the learnt causal structure, VCD is able to identify the sparse mechanism changes in the environment and re-learn *only* the intervened mechanisms. This enables fast and modular adaptation to changes in dynamics.

## 2 RELATED WORK

Predictive models of the environment can be used to derive exploration- (Sekar et al., 2020) or reward-driven (Ha and Schmidhuber, 2018; Hafner et al., 2019a;b) behaviours. In this paper, we focus on the learning of latent dynamics models. World models (Ha and Schmidhuber, 2018) train a representation encoder and a RNN-based transition model in a two-stage process. Other approaches (Hafner et al., 2019b; Zhang et al., 2019; Watter et al., 2015) learn a generative model by jointly training the representation and the transition via variational inference. PlaNet (Hafner et al., 2019b) parameterises the transition model with RNNs. E2C (Watter et al., 2015; Banijamali et al., 2018) and SOLAR (Zhang et al., 2019) use locally-linear transition models, arguing that including constraints in the dynamics model yields structured latent spaces that are suitable for control. Other approaches such as (Goyal et al., 2021b; Becker-Ehmck et al., 2019) consider latent transition as discrete mechanisms. In a similar vein, the use of latent prediction models have also been explored in the context of video prediction (Villegas et al., 2019; Denton and Fergus, 2018; Assouel et al., 2022). Our proposed approach shares the general principle that latent representations can be shaped by structured transition mechanisms (Ahuja et al., 2021). However, to the best of our knowledge, VCD is the first approach that implements a causal transition model with high-dimensional inputs.

Causal discovery methods enable the learning of causal structure from data. Approaches can be categorised as constraint-based (e.g. (Spirtes et al., 2000)) and score-based (e.g. (Hauser and Bühlmann, 2012)). The reader is referred to (Peters et al., 2017) for a detailed review of causal discovery methods. Motivated by the fact that these methods require access to abstract causal variables, recent efforts have been made to reconcile machine learning, which has the ability to operate on low-level data, and causality (Schölkopf et al., 2021). Recent advances in this area include theoretical works exploring the conditions under which disentanglement of representation is possible (Yao et al., 2022;

Hyvarinen et al., 2019; Lachapelle et al., 2021; Lippe et al., 2022a;b; Brehmer et al., 2022). Our current work situates within this broader context of causal representation learning, and aims to identify causal representations via the discovery of causal transition dynamics. (Lachapelle et al., 2021) discusses the theoretical identifiability of a latent state-space model similar to ours. Whilst we do not make identifiability claims, our work builds on the inductive biases developed in (Lachapelle et al., 2021). To the best of our knowledge, VCD is the first model that focuses on the adaptation capabilities of causal models and empirically shows their applicability to image observations.

There has been significant interest in leveraging causal reasoning in world model learning (Li et al., 2020a;b). These approaches typically rely on predefined representations such as keypoints (Li et al., 2020b) or object slots (Li et al., 2020a), and attribute changes in the dynamics to unobserved confounders, which can be estimated from observations. Our approach differs in that we couple the learning of the representation and the model, which enables dynamics-aware discovery of causal representations. Similar approaches of jointly learning representation and model structure have also been explored in different contexts such as reinforcement learning (Huang et al., 2022) and temporally intervened sequences (Lippe et al., 2022c). While the latter shares the general approach of learning sparse graphs in the latent space, it considers interventions as direct changes to the causal variables in each time step, whereas VCD formulates interventions as shifts in the dynamics *across environments*, which enables the capability of domain adaptation.

Another branch of causality-inspired work leverages the invariance of causal mechanisms by learning invariant predictors across environments (Tian and Pearl, 2001; Schölkopf et al., 2012; Peters et al., 2016; Zhang et al., 2017; Arjovsky et al., 2019). This invariance has been studied in the context of state abstractions in MDPs (Zhang et al., 2020), and imitation learning from different environments (Bica et al., 2021). In contrast, our approach models the full generative process of the data across different environments rather than learning discriminative predictors.

## 3 BACKGROUND

This section describes the prerequisite definitions and formulations for our proposed method. We highlight the strengths and weaknesses of latent state-space models, causal models and causal discovery methods, and motivate our approach that builds upon these to learn causal world models.

**Latent state-space models** In a complex environment with high-dimensional observations, such as images, learning a compact latent state space that captures the dynamics of the environment has been shown to be more computationally efficient than learning predictions directly in the observation space (Buesing et al., 2018). Given a dataset $\{(o^{0:T}, a_i^{0:T})\}_{i=0}^N$, with observations $o^t$ and actions $a^t$ at timestep $t$, a generative model of the observations can be defined using latent states $z^{0:T}$ as

$$p(o^{0:T}, a^{0:T}) = \int \prod_{t=0}^T p_\theta(o^t|z^t)p(a^t|z^t)p_\theta(z^t|z^{t-1}, a^{t-1})dz^{0:T}, \tag{1}$$

where $p_\theta(o^t|z^t)$ and $p_\theta(z^t|z^{t-1}, a^{t-1})$ are the observation model and the transition model respectively. For simplicity, we do not learn the policy term $p(a^t|z^t)$ and omit it throughout this paper as it is constant with respect to the parameter $\theta$. The variational evidence lower bound can be written as

$$\mathbb{ELBO}(\theta, \phi) = \sum_{t=0}^T \mathbb{E}_{q_\phi(z^t|o^t)}\big[log(p_\theta(o^t|z^t))\big] - \mathbb{E}_{q_\phi(z^{t-1}|o^{t-1})}\big[\mathbb{KL}[q_\phi(z^t|o^t)||p_\theta(z^t|z^{t-1}, a^{t-1})]\big],$$
$$\tag{2}$$

where $q_\phi(z^t|o^t)$ is a learnable approximate posterior. See Appendix A for the derivation. RSSM (Hafner et al., 2019b) parameterise the transition model as a feed-forward recurrent neural network.

Despite their success, these models cannot reason about changes in the environment. Specifically, they are unable to structurally utilise prior knowledge from different environments under distribution shift. To this end, we argue that imposing a *causal* structure on the transition model equips the learning agent with the ability to adapt to changes in a modular and efficient manner.

**Causal graphical models** A causal graphical model (CGM) (Peters et al., 2017) is defined as a set of random variables $\{X_1, ..., X_d\}$, their joint distribution $P_X$, and a directed acyclic graph

(DAG), $\mathcal{G} = (X, E)$, where each edge $(i, j) \in E$ implies that $X_i$ is a direct cause of $X_j$. The joint distribution admits a causal factorisation such that

$$p(x_1, ..., x_d) = \prod_{i=0}^{d} p(x_i | PA_i), \tag{3}$$

where $PA_i$ is the set of parents to the variable $X_i$ in the graph. Each of the conditional distributions can be considered as an independent *causal mechanism*.

In contrast to standard graphical models, CGMs support the notion of *interventions*, i.e., local changes in the causal distribution. Formally, an intervention on the variable $X_i$ is modelled as replacing the conditional distribution $p(x_i | PA_i)$ while leaving the other mechanisms unchanged. Given the set of intervention targets $I \subset X$, the interventional distribution can be written as

$$p'(x_1, ..., x_d) = \prod_{i \notin I} p(x_i | PA_i) \prod_{i \in I} p'(x_i | PA_i), \tag{4}$$

where $p'(\cdot | \cdot)$ is the new conditional distribution corresponding to the intervention. The SMS hypothesis (Schölkopf et al., 2021) posits that naturally occurring distribution shifts tend to correspond to sparse changes in a causal model when factorized as (3), i.e., changes of a few mechanisms only. Causal mechanisms thus tend to be *invariant* across environments (Schölkopf et al., 2012; Peters et al., 2016; Zhang et al., 2017). In this light, we argue that a *causal* world model can structurally leverage the invariance within distribution shifts as an inductive prior. In order to learn a causal model in the context of world models, we draw inspiration from recent advances in causal discovery which aim to learn causal structures from data.

**Differentiable causal discovery** We focus on methods that formulate causal discovery as a continuous optimisation problem (Brouillard et al., 2020; Ke et al., 2020; Bengio et al., 2019) as these can be naturally incorporated into the variational inference framework. Since the causal variables are *learnt* in our model, the causal discovery module is required to learn causal graphs from unknown intervention targets. In this work, we follow the formulation in Differentiable Causal Discovery with Interventional data (Brouillard et al., 2020) (DCDI), which optimises a continuously parameterised probabilistic belief over graph structures and intervention targets. See Appendix B for further detail.

In the context of learning world models from high-dimensional observations, the drawback of this approach, and of causal discovery methods in general, is that it requires access to semantically meaningful causal variables, much like classical AI required symbols in terms of which algorithms could be formulated. In the next section, we present a method to perform causal discovery and learn causally meaningful representations jointly.

## 4 VARIATIONAL CAUSAL DYNAMICS

Similar to causal discovery with interventional data, variational causal dynamics (VCD) learns from action-observation sequences from an undisturbed environment, $(o_{(0)}^{0:T}, a_{(0)}^{o:T})$, and $K$ intervened environments, $\{(o_{(k)}^{0:T}, a_{(k)}^{0:T})\}_{k=1}^{K}$. Throughout this paper, we assume that 1) changes between the environments are due to 'soft' interventions where the structure of the causal graphs remains the same but individual parameters of the mechanisms change, 2) there are no instantaneous causal dependencies in each timestep[1], and 3) the observation function does not change across environments.

The general approach of VCD follows the latent state-space model framework (Fig.2a), where a latent representation of the observations is jointly learnt with a transition model by maximising the ELBO. In contrast to existing approaches that parameterise the transition probability $p(z^t | z^{t-1}, a^{t-1})$ as a feedforward neural network (Fig.2b), VCD learns a *causal* transition model by utilising inductive biases from causal discovery. Importantly, our approach is grounded in the hypothesis that causal structures can only be discovered on semantically meaningful causal representations of the system. We therefore argue that training a representation and a transition model jointly to optimise a causal discovery objective can lead to a latent representation that affords causal transition models and is therefore semantically meaningful.[2] Taking the view that changes in the dy-

---

[1]This assumption can be readily relaxed as DCDI can handle instantaneous edges.

[2]Whilst we do not make theoretical identifiability claims for our model, the identifiability gaurantees of a similar state-space model is discussed in (Lachapelle et al., 2021).

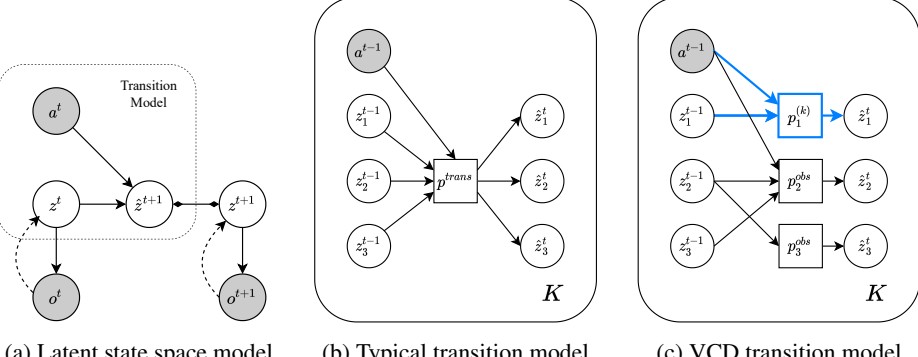

(a) Latent state space model     (b) Typical transition model     (c) VCD transition model

Figure 2: (a) The structure of latent state space models in one timestep. (b) The structure of latent transition models where the probability distribution is modelled as a fully connected neural network. (c) The causal transition model in VCD. Each individual variable has a separate conditional distribution $p_i$. Each mechanism is conditioned on a subset of the previous state and action. The blue lines highlight the intervened mechanism which is specific to the environment $k$, as opposed to the mechanisms corresponding to the black lines, which are shared across environments.

namics can be attributed to sparse causal interventions, we posit that a causally factorised transition model facilitates modular adaptation to new environments by explicitly leveraging the invariance of causal mechanisms. We propose an algorithm for such adaptation in Section 4.4.

## 4.1 CAUSAL TRANSITION MODEL

In order to perform causal discovery on the transition dynamics, the transition model in VCD is designed to mimic the structure of a CGM (Eq. 4). In the following paragraphs, we motivate and describe the three main architectural features of VCD that enable causal discovery: *independent mechanisms*, *sparse causal dependencies* and *sparse interventions*. Access to the causal graph $\mathcal{G}$ and the intervention targets for each environment $\mathcal{I}_k$ is assumed throughout this subsection. We describe the method of learning these jointly with the model in Section 4.3.

**Independent mechanisms** In contrast to a fully-connected transition model, the transition probability is factorised into individual conditional distribution as

$$p(z^t|z^{t-1}, a^{t-1}) = \prod_i^d p_i(z_i^t|z^{t-1}, a^{t-1}), \tag{5}$$

where $d$ is the dimension of the latent space, to be set as a hyperparameter. $z_i$ denotes the $i$th dimension of the latent state, and each conditional distribution $p_i$ is a one-dimensional normal distribution with mean and variance given by *separate* neural networks. This separation of parameters is motivated by the Independent Causal Mechanism principle, which states that the causal generative process of a system is composed of modules that do not inform each other (Schölkopf et al., 2021). This explicit modularity of the model structure enables the notion of interventions, where individual conditional distributions are locally changed without affecting the other mechanisms.

**Sparse causal dependencies** Following the structure of a CGM, we condition each variable only on its causal parents according to the learnable causal graph $\mathcal{G}$, rather than the full state. Given a graph $\mathcal{G}$, we define the binary adjacency mask $M^{\mathcal{G}}$ where the entry $M_{ij}^{\mathcal{G}}$ is 1 if and only if $[z^{t-1}, a^{t-1}]_i$ is a causal parent of $z_j^t$. This is consistent with the intuition that, in physical systems, states interact with each other in a sparse manner (Goyal et al., 2021a), and actions tend to have a direct effect on only a subset of the states. Under this parameterisation, the causal transition probability can be written as

$$p(z^t|z^{t-1}, a^{t-1}) = \prod_i^d p_i(z_i^t|M_i^{\mathcal{G}} \odot [z^{t-1}, a^{t-1}]), \tag{6}$$

where $\odot$ denotes the element-wise product, $[\cdot, \cdot]$ denotes the concatenation of vectors, and $M_i^{\mathcal{G}}$ is the binary mask that selects only the causal parents of $z_i$ under the graph $\mathcal{G}$.

**Sparse interventions**  Following the SMS hypothesis (Schölkopf et al., 2021), we assume that changes in distributions across the $K$ intervened environments are due to sparse interventions in the ground truth causal generative process. In order to incorporate sparse interventions in VCD, given the set of learnt intervention targets $\mathcal{I}_k$ for each environment, we define the binary intervention mask $R^{\mathcal{I}}$ where the entry $R_{ki}^{\mathcal{I}}$ is 1 if and only if the variable $z_i$ is in the set of intervention targets in environment $k$. For each variable $z_i$, $R_{ki}^{\mathcal{I}}$ acts as a switch between reusing a shared observational model and an environment-specific interventional model. The full interventional causal model of the transition probability in the environment $k$ can be written as

$$p^k(z^t|z^{t-1}, a^{t-1}) = \prod_i^d p_i^{(0)}(z_i^t|M_i^{\mathcal{G}} \odot [z^{t-1}, a^{t-1}])^{1-R_{ki}^{\mathcal{I}}} p_i^{(k)}(z_i^t|M_i^{\mathcal{G}} \odot [z^{t-1}, a^{t-1}])^{R_{ki}^{\mathcal{I}}}, \quad (7)$$

where $p^{(0)}$ is the observational mechanism that is shared across all environments and $p^{(k)}$ is the intervened distribution specific to environment $k$. Intuitively, this model ensures that all environments reuse the same conditional distributions $p^0$ unless it is deemed that a particular mechanism has been intervened on. This modular adaptation between environments facilitates structural transfer of knowledge between different environments. Note that the structure is that of a causal graphical model. The overall architecture of the VCD transition model is summarised in Fig. 2c.

## 4.2 RECURRENT MODULE

Similar to RSSM (Hafner et al., 2019b), we augment the model with a deterministic recurrent path to enable long-term predictions. To ensure that each conditional distribution only has access to the causal parents, in the same way that each conditional distribution is modelled by a separate network, each conditional distribution keeps a separate recurrent unit and a corresponding hidden activation:

$$z_i^t \sim p_i(z_i^t|h_i^t), \quad h_i^t = f_i(h_i^{t-1}, M_i^{\mathcal{G}} \odot [z_i^{t-1}, a_i^{t-1}]), \quad (8)$$

where $f_i$ is a recurrent module specific to the variable $z_i$, instantiated as a GRU (Cho et al., 2014).

## 4.3 TRAINING

The task of model training is to determine the model parameters $\theta$, the approximate posterior parameters $\phi$, as well as the causal graph $\mathcal{G}$ and the intervention targets $\mathcal{I}$. These can be jointly trained in a way similar to DCDI. We parameterise the belief over the causal adjacency matrix $M^{\mathcal{G}}$ as a random binary matrix. Each entry $M_{ij}^{\mathcal{G}}$ follows a Bernoulli distribution with success probability $\sigma(\alpha_{ij})$, where $\alpha_{ij}$ is a scalar parameter and $\sigma(\cdot)$ is the sigmoid function. Similarly, a random binary matrix $R^{\mathcal{I}}$ is parameterised using the scalar variable $\beta_{ki}$ for each entry. Unlike DCDI, due to the existence of latent variables, we cannot directly maximise the data likelihood. Instead, we maximise the expected ELBO across all environments over causal graphs and intervention masks,

$$L(\theta, \phi, \alpha, \beta) = \sum_{k \in [0,1,...,K]} \mathbb{E}_{\mathcal{G},\mathcal{I}}\big[\mathbb{ELBO}(o_{(k)}^{0:T}, a_{(k)}^{0:T}; \theta, \phi, \mathcal{G}, \mathcal{I}) - \lambda_G|\mathcal{G}| - \lambda_I|\mathcal{I}|\big], \quad (9)$$

where the ELBO term is given by the expression in Equation (2), in which the transition model, $p_\theta^{(k)}(z^t|z^{t-1}, a^{t-1})$, is further factorised as in Equation (7). The gradients through the outer expectation and the expectation term in ELBO are estimated using reparameterisation tricks (Kingma and Welling, 2014; Jang et al., 2017). For further implementation details, derivation of the lower bound, and model architectures, see Appendices A and B.

## 4.4 ADAPTATION

Due to the modular nature of the transition model, VCD can naturally adapt to new, unseen environments by jointly inferring the intervention targets and the new model parameters for the intervened mechanisms. Specifically, the transition model in a new environment can be written as

$$p^{new}(z^t|z^{t-1}, a^{t-1}) = \prod_i p_i^{(0)}(z_i^t|M_i^{\mathcal{G}} \odot [z^{t-1}, a^{t-1}])^{1-R_i'} p_i'(z_i^t|M_i^{\mathcal{G}} \odot [z^{t-1}, a^{t-1}])^{R_i'}, \quad (10)$$

where $R'$ is the intervention mask for the new environment and $p'_i$ is the environment-specific distribution. One way to implement modular transfer is to train VCD on trajectories in the new environment while fixing the trained parameters for the causal graph and the mechanisms in the undisturbed environment. The new parameters can be jointly trained in an analogous way to Eq. 9. Under this framework, VCD has the capability to identify the sparse mechanism changes in a new environment and learn the transition mechanisms only for the intervened variables. Following the SMS hypothesis, only a small subset of the mechanisms need to be adapted in a new environment. Hence, by leveraging past experience, VCD can adapt quickly to environment change. In Section 5, we show that VCD can identify sparse interventions from a small amount of data and is able to refit the world model with less data than baseline approaches.

## 5 EXPERIMENTS

In this section, we demonstrate that VCD is able to learn from multiple environments with different dynamics by learning a causal world model that explicitly captures the changes between the environments as interventions. We evaluate, qualitatively and quantitatively, the learnt representations and causal structures and show that VCD is able to capture the dynamics of the scene using sparse dependencies between states, as well as identify sparse changes between the environments. Moreover, we illustrate that, in an unseen environment, VCD can leverage past experience to perform modular adaptation, resulting in significantly improved data efficiency over the baselines.

**Dataset**  We evaluate VCD on a simulated dataset of a 2-D multi-body system which contains four particles that affect each other via a spring or an electrostatic-like force. The environment is designed such that there is an unambiguous ground-truth causal graph between the causal variables and well-defined changes in the dynamics. We consider changes such as strengthening or removing one of the springs, increasing or decreasing the mass of a particle, or constraining the position of a particle along the $x$ or the $y$ axis. Each of these changes can be considered as an intervention on one or more causal variables. We evaluate VCD in two experiments: **Mixed-state**, where the observation is given by applying an affine transformation to the ground-truth positions of the particles, and **Image**, where the observation is given by rendering the environment as a RGB image.

**Baselines**  We compare the performance of VCD against RSSM (Hafner et al., 2019b), a state-of-the-art latent world model that served as inspiration for VCD. As RSSM does not support learning from multiple environments, we consider two adaptations of RSSM with different levels of knowledge transfer between environments: (1) **RSSM**, where one transition model is trained over all environments, i.e., maximum parameter sharing across environments; and (2) **MultiRSSM**, where individual transition models are trained on each environment. This corresponds to the case where no knowledge about dynamics is transferred, i.e., each model is a local expert. We hypothesise that, compared to these two extremes of knowledge sharing, VCD is able to capture environment-specific behaviours whilst reusing invariant mechanisms via modular transfer.

**Prediction performance**  VCD and the baselines are trained on a dataset composed of 2000 trajectories from each of the undisturbed environment and five intervened environments. The models are evaluated on trajectories from a validation set drawn from the training environments. Fig. 3 shows the rollout error for each of the models with state and image observations. Unlike VCD and MultiRSSM, RSSM can only capture the average dynamics of the environments due to parameter sharing. As such, it is not able to capture environment-specific behaviours. This is reflected in the prediction accuracy of the model. In the mixed-state experiment, VCD is able to identify changes between the environments and performs as well as MultiRSSM in terms of prediction error. A similar trend is shown in the image space, where VCD outperforms the baselines.[3] See Appendix D for image rollout examples that demonstrate the difference in behaviour between RSSM and VCD.

---

[3]We note that in the image space, MultiRSSM does not perform well compared to RSSM and VCD. We hypothesise that this is due to the fact that the environment-specific transition models are only trained on data from one environment each, compared to RSSM which has access to data from all environments. VCD does not suffer from this due to modular parameter sharing.

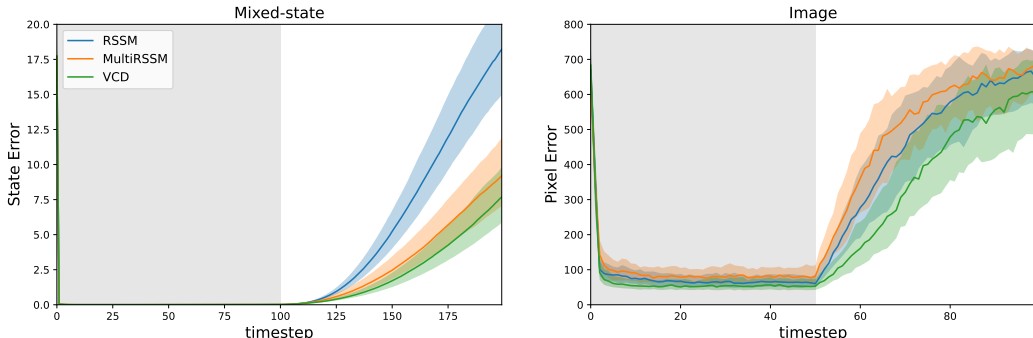

Figure 3: The rollout error measured on validation trajectories. The models receive observations on each timestep for the first half of each trajectory (shaded), after which the model is rolled out based on latent predictions. The reported error in the mixed-state environment (left) and the image environment (right) are squared error in the ground-truth state space and squared error in pixel values respectively. VCD outperforms the baselines in both modes of observation. Evaluations on the image experiments using non-pixel-based errors are available in Appendix F.

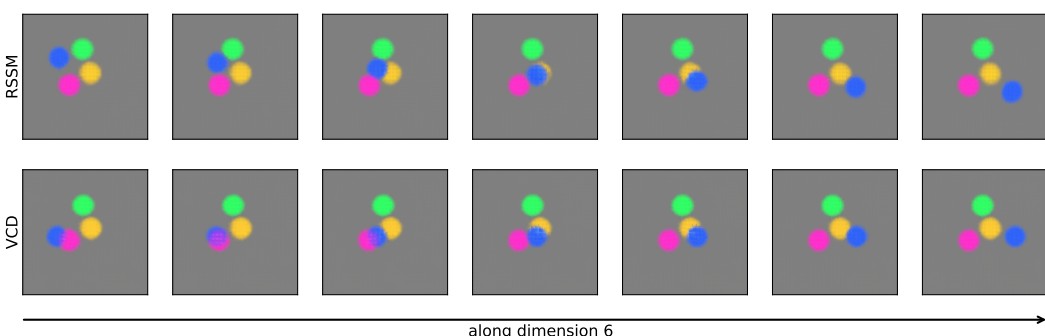

Figure 4: Reconstructed images from points sampled along dimension 6 of the latent space. Compared to the latent space learnt in RSSM, VCD is able to learn an axis-aligned representation where the blue particle moves horizontally. See Appendix D for more examples.

**Causal discovery**    One of the key hypotheses of this work is that jointly learning a representation and a transition model using causal discovery leads to causally meaningful representations. Here we examine the quality of the learnt latent space and the causal structure of the transition model. Fig. 4 shows the image reconstructions of points drawn from a straight line along dimension 6 of the latent space. Since both models are initialised with the same encoder, this provides a qualitative intuition as to how the causal discovery inductive bias shapes the latent space. Whilst this dimension of the latent space in RSSM is able to encode only the blue particle, VCD is able to learn an *axis-aligned* coordinate ($x$ coordinate) of the blue particle with respect to the environment. Note that the motion of bouncing off the boundaries is only separable in the $x, y$ frame, implying that the dynamics in axis-aligned coordinates is sparser. We present a further analysis of the learnt causal representation in Appendix D. As desired, the learnt causal graphs in both the mixed-state and the image environment are found to be sparse.

**Adaptation**    We provide empirical evidence that VCD can adapt to a new environment with less data compared to RSSM and MultiRSSM by reusing learnt mechanisms in a modular fashion. We collect datasets of different sizes in a previously unseen intervened environment where particle 1 is constrained horizontally. RSSM and MultiRSSM adapt to the new environment by optimizing the ELBO (Eq. 2), with the difference that MultiRSSM instantiates a new transition model randomly and RSSM initialises the transition model using pre-trained parameters. VCD performs adaptation by jointly estimating the intervention targets and the parameters of the intervened mechanisms. The models are evaluated on a validation set of trajectories in the new environment. Fig. 5 shows the

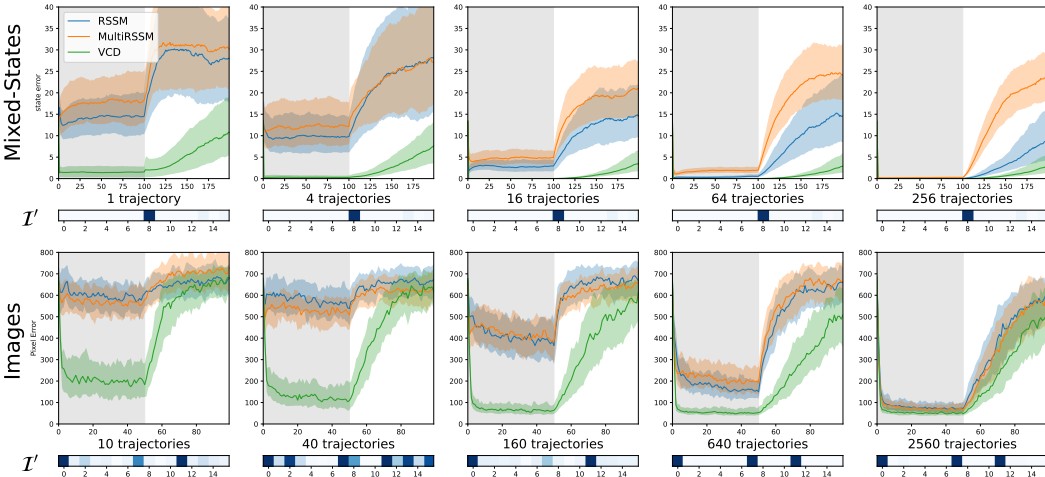

Figure 5: Rollout errors in the mixed-state and image experiments where the models are trained on datasets of varying sizes. The models receive observations for the first half of the trajectory (shaded), and perform latent space prediction for the rest. VCD significantly outperforms the other models with little data. The bar below each plot shows the learnt intervention targets in the new environment. VCD is able to reuse most of the previously learnt mechanisms, as indicated by the sparsity in the learnt intervention targets. In the mixed-state, we verify that the learnt intervention target (dimension 8) corresponds to the ground-truth target ($y_1$), see Appendix D. Evaluations on the image experiments using non-pixel-based errors are available in Appendix F.

rollout error of the models trained on datasets of varying size. Across all models, performance improves as the dataset grows. However, in contrast to RSSM and MultiRSSM, which overfits the dataset when the number of trajectories is small, VCD is able to predict significantly more accurately. This is because VCD estimates the intervention targets and reuses trained modules that remain invariant. Remarkably, in the mixed-state experiment, VCD is able to converge to a single latent dimension that encodes the $y$ position of particle 1 with just one training trajectory. See Appendix E for more experiments with different interventions.

## 6 CONCLUSION AND DISCUSSION

In this paper, we propose VCD, a predictive world model with a causal structure that is able to consume high-dimensional observations. This is achieved by jointly training a representation and a causally structured transition model using a modified causal discovery objective. In doing so, VCD is able to identify causally meaningful representations of the observations and discover sparse relationships in the dynamics of the system. By leveraging the invariance of causal mechanisms, VCD is able to adapt to new environments efficiently by identifying relevant mechanism changes and updating in a modular way, resulting in significantly improved data efficiency.

**Limitations and future directions** Our experiments highlight VCD's ability to discover causal variables and mechanisms from image observations. Future work should investigate the efficacy of the proposed method in more challenging environments such as (Baradel et al., 2019; Ahmed et al., 2021). One possible drawback of the proposed framework is that the causal graph is assumed to be static across all timesteps and all environments. In more complex environments, it could be more difficult for the model to discover temporally sparse interactions such as collisions, and adapt to scenes with different numbers of objects. As such, exciting directions for future research also include exploring the synergy between causal world models and object-centric generative models (Wu et al., 2021; Engelcke et al., 2021; von Kügelgen et al., 2020), temporally-local causal influence detection (Seitzer et al., 2021; Pitis et al., 2020) and relational reasoning (Goyal et al., 2021b; Kipf et al., 2019).

**Reproducibility Statement**   In order to ensure reproducibility of the results, derivation of the variational lower bound, details of the model architectures used in the experiments, implementation details for the proposed algorithms, and details of the experiment setup is included in App. A, B and C. The code for the experiments as well as the data generation process will be made available for the reviewing process and we intend to release the code for the camera-ready version of the paper.

**Ethics Statement**   While the present work significantly advances the current state-of-the-art in world-modelling and causal representation learning, we expect its immediate impact outside of the machine learning community to be low as current methods can not yet deal effectively with real-world scenarios.

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

## A    DERIVATIONS

Using the approximate posterior $q(z^{0:T}|o^{0:T}, a^{0:T}) = \prod_t q(z^t|o^t)$, the variational log lower bound for the latent state-space model (Eq. 2) can be derived from importance weighting and Jensen's inequality:

$$\log p(o^{0:T}, a^{0:T}) = \log \int \prod_{t=0}^{T} p(o^t|z^t)p(a^t|z^t)p(z^t|z^{t-1}, a^{t-1})dz^{0:T} \tag{11}$$

$$= \log \int \prod_0^T p(o^t|z^t)p(a^t|z^t)\frac{p(z^t|z^{t-1}, a^{t-1})}{q(z^t|o^t)}q(z^t|o^t)dz^{0:T} \tag{12}$$

$$\geq \mathbb{E}_{\prod q(z^t|o^t)}\left[\log\left(\prod_o^T p(o^t|z^t)p(a^t|z^t)\frac{p(z^t|z^{t-1}, a^{t-1})}{q(z^t|o^t)}\right)\right] \tag{13}$$

$$= \sum_0^T \mathbb{E}_{q(z^t|o^t)}\left[\log p(o^t|z^t) + \log p(a^t|z^t)\right]$$
$$+ \mathbb{E}_{q(z^{t-1}|o^{t-1})}\left[\mathbb{E}_{q(z^t|o^t)}\left[\log p(z^t|z^{t-1}, a^{t-1}) - \log q(z^t|o^t)\right]\right] \tag{14}$$

$$= \sum_0^T \mathbb{E}_{q(z^t|o^t)}\left[\log p(o^t|z^t) + \log p(a^t|z^t)\right]$$
$$- \mathbb{E}_{q(z^{t-1}|o^{t-1})}\left[\mathbb{KL}\left[q(z^t|o^t)||p(z^t|z^{t-1}, a^{t-1})\right]\right]. \tag{15}$$

Since the policy $p(a^t|z^t)$ is constant with respect to the model parameters, we omit this term and write the ELBO objective as

$$\mathbb{ELBO}(\theta, \phi) = \sum_{t=0}^{T} \mathbb{E}_{q_\phi(z^t|o^t)}\left[log(p_\theta(o^t|z^t))\right] - \mathbb{E}_{q_\phi(z^{t-1}|o^{t-1})}\left[\mathbb{KL}[q_\phi(z^t|o^t)||p_\theta(z^t|z^{t-1}, a^{t-1})]\right], \tag{16}$$

where $\theta$ is the model parameter and $\phi$ is the parameter for the approximate posterior. $\theta$ and $\phi$ are omitted henceforth to simplify notation. In VCD, the KL divergence term can be further decomposed by exploiting the structure of the transition model (Eq. 7), and the assumption that variables within each timestep are independent:

$$\mathbb{KL}\left[q(z^t|o^t)||p^{(k)}(z^t|z^{t-1}, a^{t-1})\right] \tag{17}$$

$$= -\int \log\left(\frac{p^{(k)}(z^t|z^{t-1}, a^{t-1})}{q(z^t|o^t)}\right)q(z^t|o^t)dz^t \tag{18}$$

$$= -\sum_{i=0}^{d} \int \log\left(\frac{p_i^{(0)}(z_i^t|M_i^{\mathcal{G}} \odot [z^{t-1}, a^{t-1}])^{1-R_{ki}^{\mathcal{I}}}p_i^{(k)}(z_i^t|M_i^{\mathcal{G}} \odot [z^{t-1}, a^{t-1}])^{R_{ki}^{\mathcal{I}}}}{q(z_i^t|o^t)}\right)q(z_i^t|o^t)dz_i^t \tag{19}$$

$$= -\sum_0^d \left((1 - R_{ki}^{\mathcal{I}})\int \log\left(\frac{p_i^{(0)}(z_i^t|M_i^{\mathcal{G}} \odot [z^{t-1}, a^{t-1}]}{q(z_i^t|o^t)}\right)q(z_i^t|o^t)dz_i^t\right.$$
$$\left. + R_{ki}^{\mathcal{I}}\int \log\left(\frac{p_i^{(k)}(z_i^t|M_i^{\mathcal{G}} \odot [z^{t-1}, a^{t-1}])}{q(z_i^t|o^t)}\right)q(z_i^t|o^t)dz_i^t\right) \tag{20}$$

$$= -\sum_0^d \left((1 - R_{ki}^{\mathcal{I}})\mathbb{KL}\left[q(z_i^t|o^t)||p_i^{(0)}(z_i^t|M_i^{\mathcal{G}} \odot [z^{t-1}, a^{t-1}])\right]\right.$$
$$\left. + R_{ki}^{\mathcal{I}}\mathbb{KL}\left[q(z_i^t|o^t)||p_i^{(k)}(z_i^t|M_i^{\mathcal{G}} \odot [z^{t-1}, a^{t-1}])\right]\right). \tag{21}$$

The KL terms can be computed analytically since the conditional distributions in the last expression are univariate Gaussian distributions. In training time, the gradients through the expectation terms in the ELBO is estimated by drawing a sample from the posterior distribution using the reparameterisation trick (Kingma and Welling, 2014).

## B    IMPLEMENTATION DETAIL

### B.1    DCDI AND GRAPH LEARNING

This section covers the formulation of DCDI (Brouillard et al., 2020) and the graph learning method. These are subsequently used in the learning of VCD.

Given samples from an observed data distribution $P_X^{(0)}$ and $K$ intervened distributions $P_X^{(k)}$, DCDI optimises a probabilistic belief over causal graphs $\mathcal{G}$ and intervention targets $\mathcal{I}$. Specifically, these are encoded as random binary matrices, $M_{\mathcal{G}}$ and $R_{\mathcal{I}}$, where $M_{ij}^{\mathcal{G}} = 1$ implies that the edge $(i, j)$ is in the causal graph, and $R_{ki}^{\mathcal{I}} = 1$ implies that the variable $x_i$ is in the intervention targets in environment $k$. Each entry in $M^{\mathcal{G}}$ follows an independent Bernoulli distribution, parameterised by matrix $\alpha$ where $P(M_{ij}^{\mathcal{G}} = 1) = \sigma(\alpha_{ij})$. $R_{ki}^{\mathcal{I}}$ is similarly parameterised by $\beta$. Under this parameterisation, causal discovery can be formulated as maximising the expected data log likelihood with sparsity regularisation,

$$L(\theta, \alpha, \beta) = \mathbb{E}_{\alpha, \beta} \left[ \sum_{k=0}^{K} log[p_{\theta}^{(k)}(x_1^k, ..., x_d^k; \mathcal{G}, \mathcal{I})] - \lambda_G |\mathcal{G}| - \lambda_I |\mathcal{I}| \right], \tag{22}$$

where $p^{(k)}$ is the data likelihood under causal graph $\mathcal{G}$ and intervention targets $\mathcal{I}$ in the $k$th environment, as factorised in eq.(4). The conditional distributions are parameterised as feedforward neural networks with parameter $\theta$; $\lambda_{G,I}$ are hyperparameters to control sparsity. In the original DCDI framework, this is also subject to an acyclicity constraint. However, this is not neccessary in the context of our work as we assume there are no instantaneous causal effects (i.e., within a timestep).

The training objective for VCD can be viewed as a modified version of the DCDI objective, where the likelihood term is replaced with the ELBO (Eq. 16),

$$L^{VCD}(\theta, \phi, \alpha, \beta) = \mathbb{E}_{\alpha, \beta} \left[ \sum_{k=0}^{K} \sum_{t=0}^{T} \mathbb{ELBO}(\theta, \phi; \mathcal{G}, \mathcal{I}) - \lambda_G |\mathcal{G}| - \lambda_I |\mathcal{I}| \right], \tag{23}$$

Note that the expected number of edges in $\mathcal{G}$ and $\mathcal{I}$ given $\alpha$ and $\beta$ is simply the sum of the probability of each entry being one. Therefore, the training objective can be computed as:

$$L^{VCD}(\theta, \phi, \alpha, \beta) = \mathbb{E}_{\alpha, \beta} \left[ \sum_{k=0}^{K} \sum_{t=0}^{T} \mathbb{ELBO}(\theta, \phi; \mathcal{G}, \mathcal{I}) \right] - \lambda_G \sum_{ij} \sigma(\alpha_{ij}) - \lambda_I \sum_{ki} \sigma(\beta_{ki}). \tag{24}$$

The gradients through the outer expectation can be estimated using the Gumbel-Softmax trick (Jang et al., 2017). To implement this, the ELBO term is evaluated with a sample of the causal graph using the following expression for each entry,

$$M_{ij}^{\mathcal{G}} = \mathbb{I}(\sigma(\alpha_{ij} + L_{ij}) > 0.5) + \sigma(\alpha_{ij} + L_{ij}) - stop\_gradient(\sigma(\alpha_{ij} + L_{ij})), \tag{25}$$

where $\mathbb{I}(\cdot)$ is the indicator function, $L_{ij}$ is a sample from the logistic distribution, and $stop\_gradient$ is a function that does not change the value of the argument but sets the gradient to zero. Samples for the intervention targets are similarly acquired. Note that the sample is used throughout each trajectory, i.e. the same sample graph and intervention targets are used for all of $T$ timesteps.

## B.2 Model Architecture

**Mixed-state** In the mixed-state experiment, all conditional distributions (including encoders, decoders and transition models) are parameterised by feedforward MLPs with two hidden layers of 64 hidden units each. The recurrent modules are implemented as GRUs (Cho et al., 2014) with 64 hidden units. Distributions in the latent space are 16-dimensional diagonal Gaussian distributions with predicted mean and log variance.

**Image** In the image experiment, the encoders and decoders are parameterised as convolutional and deconvolutional networks from (Ha and Schmidhuber, 2018). In the RSSM models, the transition models are parameterised as feedforward MLPs with two hidden layers of 300 hidden units. The recurrent module is a GRU with 300 hidden units. In VCD, to compensate for the fact each dimension in the latent space has a separate model, the number of hidden units in the GRU and MLP are reduced to 32 to avoid over-parameterisation. We found that initialising the encoders and decoders by pretraining them as a variational autoencoder helped with training stability for both RSSM and VCD.

In both experiments, the training objective is maximised using the ADAM optimiser (Kingma and Ba, 2014) with learning rate $10^{-3}$ for mixed-state, and $10^{-4}$ for images. In both environments, we clip the log variance to $-3$, with a batch size of two trajectories from each of six environments with $T = 50$. In VCD, the hyperparameters $\lambda_{\mathcal{G}}, \lambda_{\mathcal{I}}$ are both set to 0.01. All models are trained on a single Nvidia Tesla V100 GPU.

## C Experiment Detail

Table 1: List of interventions

| ID | Intervention | Intervention targets |
|---|---|---|
| 1 | Remove spring between 1 and 2 | $x_1, y_1, x_2, y_2$ |
| 2 | Remove spring between 2 and 3 | $x_2, y_2, x_3, y_3$ |
| 3 | Increase mass 1 | $x_1, y_1, x_4, y_4$ |
| 4 | Increase mass 2 | $x_2, y_2$ |
| 5 | Increase mass 3 | $x_3, y_3, x_4, y_4$ |
| 6 | Decrease mass 1 | $x_1, y_1, x_4, y_4$ |
| 7 | Decrease mass 2 | $x_2, y_2$ |
| 8 | Decrease mass 3 | $x_3, y_3, x_4, y_4$ |
| 9 | Increase spring constant between 1 and 2 | $x_1, y_1, x_2, y_2$ |
| 10 | Increase spring constant between 2 and 3 | $x_2, y_2, x_3, y_3$ |
| 11 | Constrain movement of 1 to vertical only | $x_1$ |
| 12 | Constrain movement of 1 to horizontal only | $y_1$ |
| 13 | Constrain movement of 2 to vertical only | $x_2$ |
| 14 | Constrain movement of 2 to horizontal only | $y_2$ |
| 15 | Constrain movement of 3 to vertical only | $x_3$ |
| 16 | Constrain movement of 3 to horizontal only | $y_3$ |
| 17 | Constrain movement of 4 to vertical only | $x_4$ |
| 18 | Constrain movement of 4 to horizontal only | $y_4$ |

Table 2: The modified MCC scores for RSSM and VCD in both experiments

| Expriment | RSSM | VCD |
|---|---|---|
| Mixed-state | 0.728 | **0.975** |
| Image | 0.715 | **0.908** |

**Interventions** The ground truth states of the multi-body dynamics environment is the $x$ and $y$ coordinates of each particle. The full list of possible interventions on the environment is provided in Table 1. Note that the forces between particle 1, 3 and 4 are proportional to their masses. Hence intervening on the mass of particle 1 and 3 also affect the dynamics of particle 4. In the experiments, all models are trained in the undisturbed environment and intervened environments 1, 5, 11, 14, 17.

In the mixed-state experiment, the observation function is a mixing matrix where each entry is drawn from a unit Gaussian distribution. In the image experiment, the observation is given by rendering the system to a $128 \times 128 \times 3$ image. In both experiments, the models are trained on a training set of 2000 trajectories from each of the six environments and evaluated on a validation set of 400 unseen trajectories.

# D  QUALITATIVE EXPLORATION FOR LEARNT CAUSAL STRUCTURE

## D.1  ROLLOUT PREDICTION

Section 5 shows the rollout accuracy of the baselines and VCD. Here we demonstrate qualitatively that VCD is able to capture environment-specific behaviours that RSSM cannot learn due to maximal parameter sharing. Figure 6 shows sample rollouts in the image space from RSSM and VCD in intervened environment 11, i.e. the yellow particle is constrained horizontally and only moves vertically. The yellow particle in the VCD rollout stays along a vertical line whereas RSSM fails to capture this environment-specific constraint.

## D.2  REPRESENTATION QUALITY

In this subsection, we explore the quality of the learnt latent space. The key hypothesis of our work is that training the representation jointly with the transition model to maximise a causal discovery objective serves as an inductive bias that helps to structure the latent space in a causally meaningful way. For both experiments,

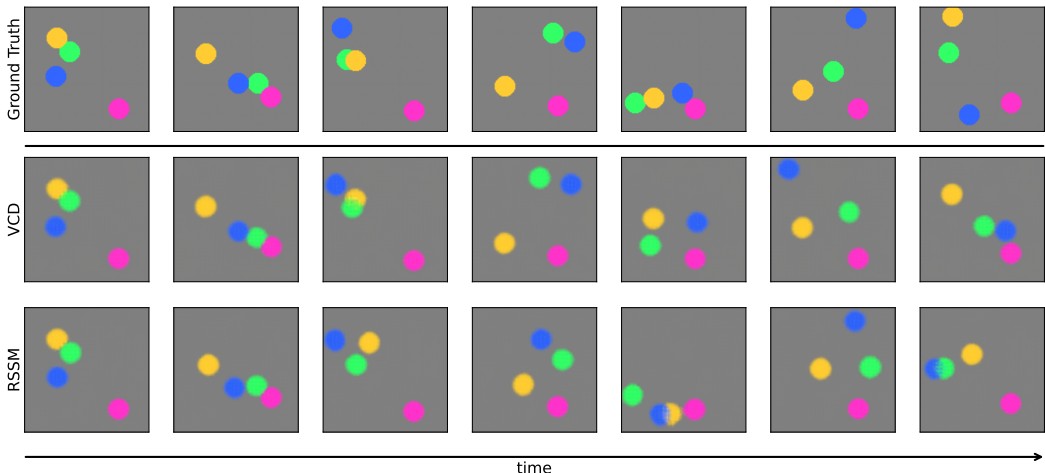

Figure 6: Sample rollouts (without observations) from an intervened environment. VCD successfully captures the constraint on the yellow particle (vertical movement only) over a long time horizon. The frames are sampled 5 timesteps apart from each other.

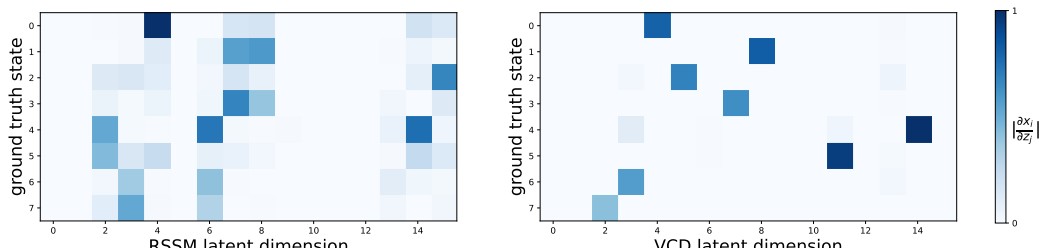

Figure 7: The average magnitude of the derivative of the ground-truth states $x_i$ w.r.t. the learnt latent states $z_j$. VCD is able to learn a latent representation where each dimension captures information about one state only, i.e. only one state variable changes when we perturb the latent representation along one dimension. On the other hand, the RSSM latent space encodes state information in a more entangled manner, i.e. multiple ground-truth states are affected when the latent representation is perturbed in one dimension.

we provide the mean absolute correlation coefficient score for the representations for both RSSM and VCD.[4] We see that in both cases, training with the sparsity regularisation in VCD achieves a higher MCC score, which confirms that VCD learns a more disentangled representation.

**Mixed-state**   Since the observation mixing function is linear and invertible in the mixed-state environment, we can directly access the level of disentanglement of the latent space with respect to the ground-truth state of the environment, i.e. the $x$ and $y$ coordinate of the particles. Fig. 7 shows the average magnitude of the entries of the Jacobian matrix between the ground-truth state and the learnt latent state. Each entry measures the changes in each ground-truth state variable when the latent representation is perturbed along each dimension. By using the discovery of causal transition models as an inductive bias, VCD is able to learn a disentangled representation where each ground-truth state is captured by only one latent dimension. In contrast, RSSM learns a representation that is not sparse.

**Image**   The quality of the learnt representation is discussed in Section 5. Here we show reconstruction samples along all dimensions of the latent space in the RSSM and VCD representation. Note that since both encoders are initialised from the same pretrained VAE, the difference in the latent space arise because of the causal discovery objective in VCD. Fig. 12 shows the changes in the reconstruction when the RSSM repre-

---

[4]Due to the fact that the ground-truth state dimension (8) is less than the latent state dimension (16), we modify the MCC score by calculating the mean of the top 8 scores out of the 16, effectively ignoring the latent variables that do not capture the state information.

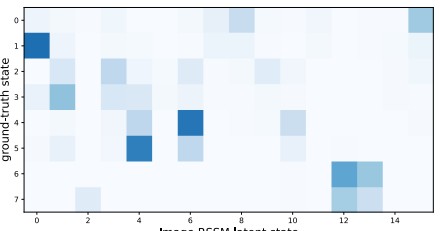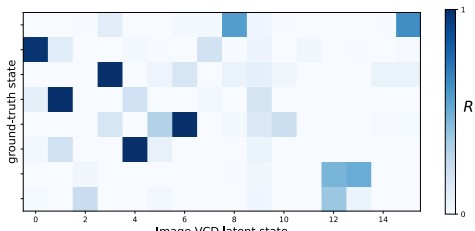

Figure 8: The $R^2$ of a linear regression between the ground-truth states and the learnt latent variables. This shows a similar pattern as the mixed-state experiments where the VCD representation in general only capture information about one ground-truth state.

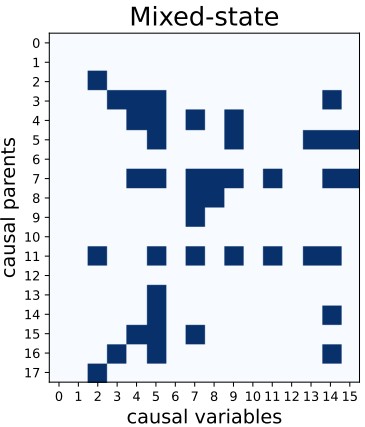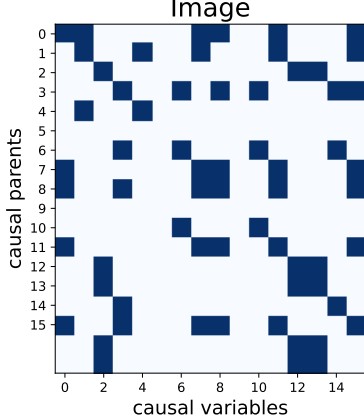

Figure 9: The learnt causal graphs for the mixed-state and image experiments, obtained by binarising the learnt edge probabilities such that a blue square at $(i, j)$ implies $\sigma(\alpha_{ij}) > 0.5$, i.e. $i$ is a causal parent of $j$.

sentation is perturbed in each dimension of the latent space. Fig. 11 shows the same for VCD. As discussed in the main text, VCD learns a *axis-aligned* representation that affords a sparse causal graph. For example, dimension 1 and 3 captures the $y$ and $x$ coordinates of the green particle respectively. In contrast, RSSM learns a representation that is not axis-aligned. Fig. 8 plots the $R^2$ linear regression scores between the learnt latent variables and the ground-truth variables, showing that the VCD representation can in general disentangle individual states, as indicated by the more salient squares.

### D.3 LEARNT CAUSAL GRAPHS AND INTERVENTION TARGETS

In this subsection, we explore the quality of the learnt causal graph and intervention targets. Fig. 9 shows the learnt causal graphs for the mixed-state and image experiments. Fig. 10 shows the learnt intervention targets for each environment. The sparsity of the learnt graph and targets is summarised in Table 3. VCD has identified 42

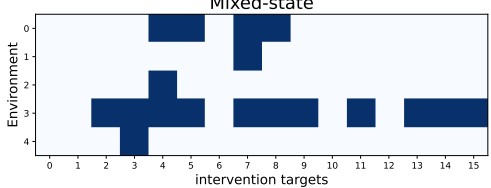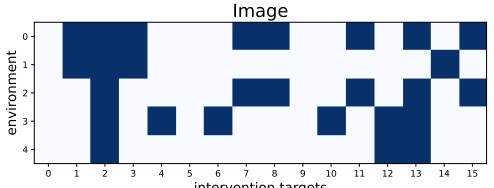

Figure 10: The learnt intervention targets for the mixed-state and image experiments, obtained by binarising the learnt probabilities such that a blue square at $(k, i)$ implies $\sigma(\beta_{ki}) > 0.5$, i.e. $i$ is an intervention target in environment $k$.

Table 3: Sparsity of learnt causal graphs and intervention targets. We compare the learnt graph with the ground truth causal graph by mapping each latent dimension to a ground-truth state as shown in Fig. 7 and Fig. 8. Note that the number of correct edges and the false positives do not sum to the number of edges in the graph because the edges corresponding to 'dummy' variables are ignored.

| Observation | | # of edges | Correct Edges | Missed Edges | False Positives |
|---|---|---|---|---|---|
| Mixed state | causal graph | 42/288 | 19 | 7 | 8 |
| | intervention targets | 18/80 | 10 | 0 | 5 |
| Image | causal graph | 73/288 | 22 | 4 | 13 |
| | intervention targets | 31/80 | 10 | 0 | 17 |

and 73 causal edges in the mixed-state and image experiments respectively, out of 288 possible edges. Viewed in conjunction with the prediction performance results, this shows that VCD is able to learn a world model that is sparsely connected and affords modular parameter sharing *without* compromising on prediction accuracy.

**Mixed-state** In the mixed-state experiments, each ground-truth state can be mapped to a latent dimension using the average magnitude Jacobian matrix (Fig. 7). We use this mapping to compare the learnt causal graph with the ground truth causal structure of the environment and report the number of correctly identified edges. Table 3 summarises the quality of the learnt graph. VCD is able to identify a majority of the correct causal dependencies and *all* of the intervention targets. Upon further inspection of the learnt causal graph, we find that all of the seven missed edges correspond to the $1/||\delta\mathbf{x}||^2$ terms that scale the electrostatic-like forces. We hypothesise that the model cannot capture these dependencies as they are not as significant as the other forces.

**Image** In the image space, a similar comparison is made using the $R^2$ scores. Each ground-truth state is mapped to a latent dimension by choosing the dimension with the highest $R^2$ score. This leaves unused latent states, which are ignored in the analysis provided in Table 3. We also qualitatively explore the learnt causal relationships. Focusing on dimension 3, for example, which encodes the $x$ coordinate of the green particle, the ground-truth causal parents of this variable is the $x$ coordinates of the yellow and the blue particles. In the learnt causal graph, on column three, the learnt causal parents are dimensions 3, 6, 8, 14 and 15. The visualisation of the latent space (Fig. 11) suggests that dimensions 6 and 8 captures the $x$ coordinates of the yellow and blue particles respectively, meaning that VCD has learnt the correct causal parents.

A similar analysis can be carried out on the intervention targets. Focusing on intervened environment 0, for example, the learnt intervention targets are dimension 1, 2, 3, 7, 8, 11, 13, 15. The ground-truth intervention ID is 1, i.e. the intervention targets are $x, y$ coordinates of the yellow and green particles (see. table 1). By inspecting the VCD latent space (Fig. 11), dimensions 1 and 3 encodes the position of the green particle and dimensions 7, 8 and 11 encodes the position of the yellow particle. This shows that VCD is able to identify the changes in the environments.

In summary, while VCD identifies some false positive edges, it is able to capture the causal parents and the intervention targets in each environment.

## E    FURTHER EXPERIMENTS

This section provides extra adaptation experiment results similar to Fig. 5 in the main text, where the models adapt to intervention number 12 (see table 1). We present experiment results for adaptation to different types of interventions (intervention numbers 4, 9, 13). Fig 13, 14 and 15 show the adaptation plots. The results exhibit a consistent pattern where VCD significantly outperforms the baselines in the low data regime by identifying sparse mechanism changes.

## F    NON-PIXEL-BASED ERROR

In the image experiments, while pixel error is indicative of model performance in the short term, it provides limited clarity in long term predictions as two non-overlapping balls leads to the same pixel error regardless of their distance. In this section, we present an alternative latent space distance-based evaluation metric, Hit at 5 (H@5). This metric is defined by the proportion of episodes where the predicted latent representation is in the top-5 nearest neighbour set of the encoded ground-truth image. A H@5 score of 1 means all predicted latent states lie within a close distance to the ground-truth representation. Fig 16 and 17 shows the prediction errors

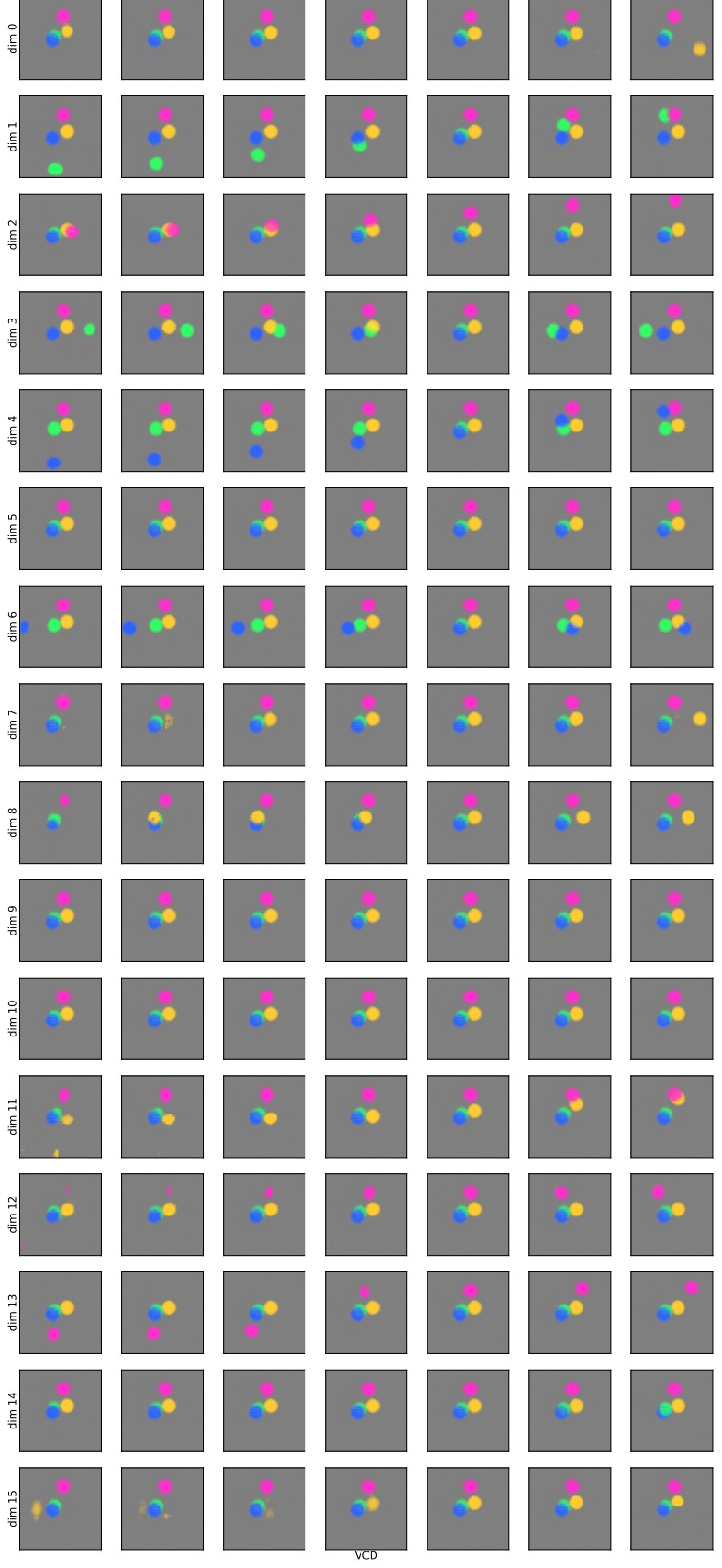

Figure 11: Visualisation of the learnt latent space in VCD.

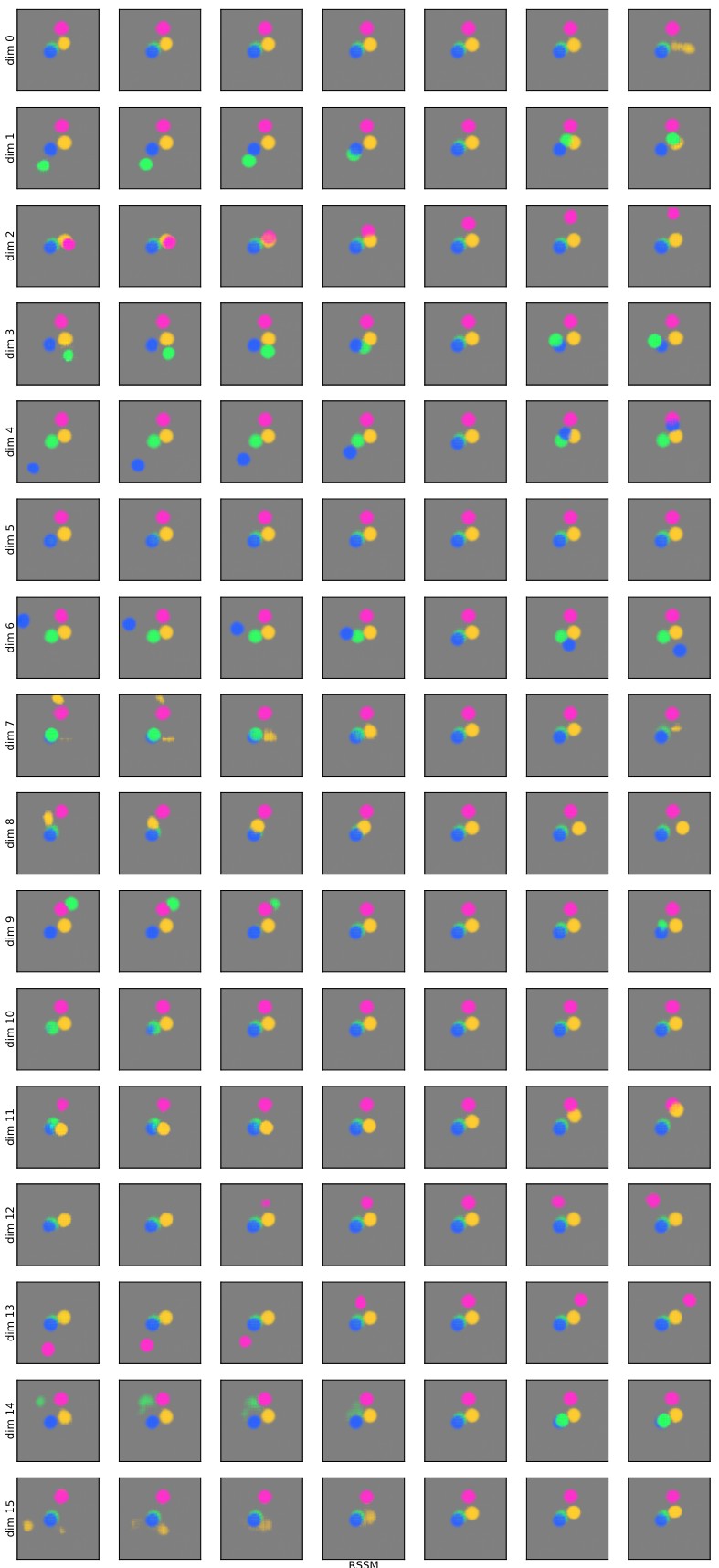

Figure 12: Visualisation of the learnt latent space in RSSM.

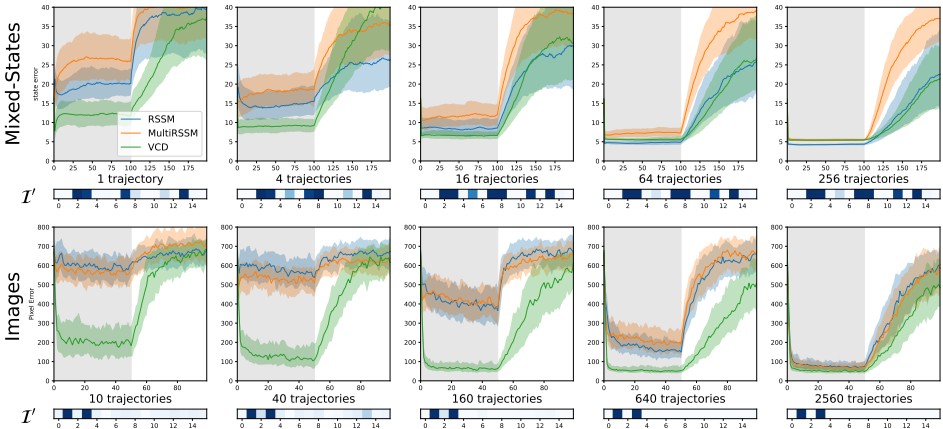

Figure 13: Adaptation results for intervention 4.

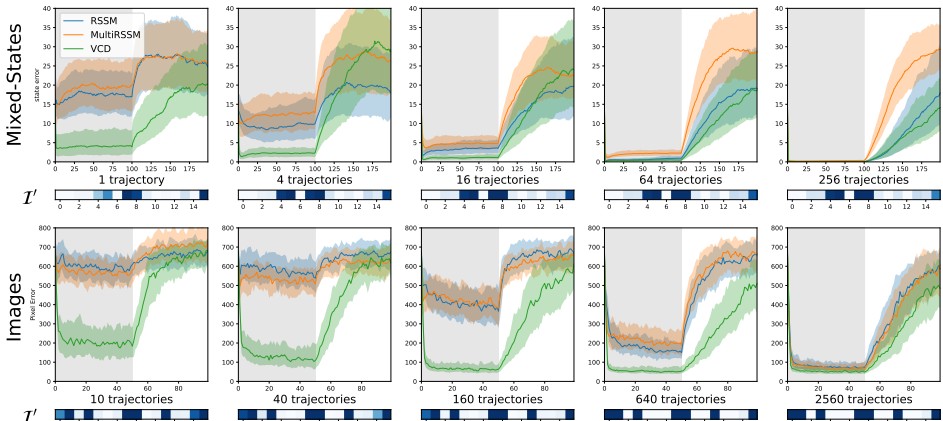

Figure 14: Adaptation results for intervention 9.

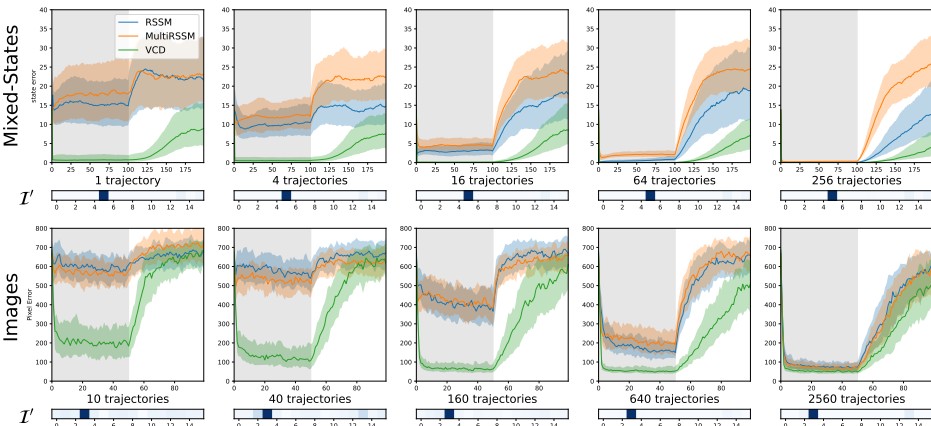

Figure 15: Adaptation results for intervention 13.

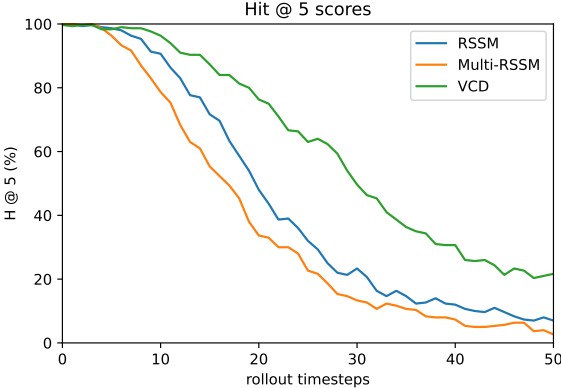

Figure 16: The H@5 score (higher is better) over rollout timesteps, i.e. unshaded regions in the main plots. This shows a consistent result as the main text, where VCD outperforms the baselines.

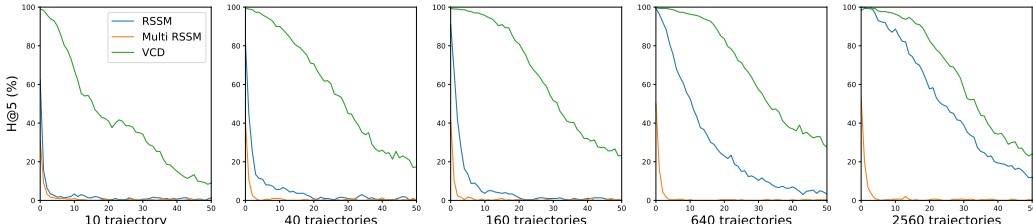

Figure 17: The adaptation plots with the H@5 scores.

and the adaptation plots respectively. These are consistent with the results shown in the main text, where VCD outperforms the baselines in prediction accuracy as well as adaptation speed.

