# OpenReview forum: "Variational Causal Dynamics: Discovering Modular World Models from Interventions"
_ICLR.cc/2023/Conference — Submitted to ICLR 2023_

### Official Review · Reviewer_Tfgu · 2022-10-21

**Confidence:** 4
**Correctness:** 4
**Technical Novelty And Significance:** 3
**Empirical Novelty And Significance:** 2
**Recommendation:** 6

**Clarity, Quality, Novelty And Reproducibility:**

The paper is very clearly written, the methods are natural given the setup. The method is complex because there are a lot of moving pieces, but I think each part is relatively straightforward to reproduce. The authors provided code to reproduce the experiments in the supplement.

Minor:
 - First page, first paragraph, this sentence is a little awkward: "This latter aspect, it is conjectured, is was...". Consider: "It is conjectured that this latter aspect is what...". Alternatively, a better rewording would cite whoever made this conjecture (or a survey) to avoid the passive voice. I.e. "Someone [2004] conjecture that this latter ....")

**Strength And Weaknesses:**

Strengths:
 - A very clear and well written paper that combines most of the key ideas from the recent progress in causal representation learning and causal discovery into a latent state-space model. None of the individual components were particularly surprising, but the combined architecture and train procedure is appealing and very well motivated.

Weakness:
 - The experiments were conducted on a very toy dataset. This style of data generating process is common in the recent papers in causal representation learning (although even there, richer datasets like 3DIdent, etc. are becoming more common), but in those papers, the experiments are primarily to ask whether identifiable models are learnable in practice (and to evaluate finite sample / optimization error / etc issues that are hidden in identifiability proofs). Because this paper isn't claiming any theoretical advances (and to be clear, I don't think it needs to!), it leans more heavily on experiments to show the value of these approaches. Given the really strong setup and motivation, I was hoping that the experiments would either provide strong evidence for the value of this approach, or alternatively show where there is need to further improvements.

   The experiments do provide evidence that there is a benefit to taking advantage of causal structure, but it is not clear whether this generalizes beyond these environments that are specifically designed to show the difference between these approaches. And given that the image based experiments showed relatively modest differences when a large number of trajectories are used (Figure 3), I'm not sure that there will be any significant different in a more natural environment (e.g. an RL-style task). In the adaption experiments with low data regimes, the differences are far more dramatic. But that raises the question: is the advantage of imposing causal structure only about sample complexity, such that if we see enough complexity, the differences are washed out?

 - Causal discovery section: I would have liked to see a quantitative metric like the mean absolute correlation coefficient to measure disentanglement. In this setting you'll need to do a matching based on covariance between the true latents and the model's latents and drop the least correlated latents (to account for differences in size). Dropping the least correlated latents will bias the scores upwards, but they will still be comparable across models that use the same architecture. I would also include a linear disentanglement score ($r^2$ of a linear regression between the true latents and the predicted latents).

 - The paper draws on a number of areas to compose its architecture  (causal discovery / identifiable representation learning /  independent mechanisms / etc), so I think the experimental section would be significantly improved if it gave guidance on what components need to improve in order to see bigger differences with RSSM and to generalize to richer environments. I.e. were do we need breakthroughs to see the biggest gain in performance? Is it causal discovery that's letting the method down? Or the encoder? Or the parameterization of the independent mechanisms? Something else?

**Summary Of The Paper:**

This paper describes variational causal dynamics (VCD), an approach that replaces a latent state-space model with a causal transition model that is learnt via a combination of differentiable causal discovery (Brouillard et al., 2020) and a set of independent deep networks that are defined the inferred causal graph. Additionally, they have a simple masking strategy to define the interventional distribution. The whole pipeline is trained via variational inference (optimizing the ELBO). They evaluate the approach of a simple physics-based task involving multiple coloured balls connected by spring forces.

**Summary Of The Review:**

Overall, I liked the paper - it proposes a natural latent variable model that takes advantage of the recent advances in our understanding of causal representation learning, but I felt like it was let down by an experimental section that didn't do a good enough job of testing the limits of the approach.

---

> ### Author Response · Authors · 2022-11-17
> **Response to reviewer Tfgu**
>
> We thank reviewer Tfgu for their time and thorough review, and are delighted to read their positive comments. Below we address the issues raised.
> - **Clarification of experimental results**: Indeed, as the reviewer suggests, we argue that the main contribution of our work lies in demonstrating the benefits of modular models in few-shot adaptation. We believe that our focus on domain adaptation has significant implications for applications such as sim-to-real transfer, where plenty of data is available in training but the amount of data is limited in the adaptation environment. To this end, the significantly improved performance in the low data regime demonstrates our claim. It is expected that, since RSSM is an expressive world model, RSSM would be able to accurately capture the dynamics of the scene given enough data. Hence, we would like to clarify that the main point of the large data experiments is to show that VCD can achieve the same level of performance as the baselines, even under the sparsity constraint which effectively reduces the capacity of the model. In fact, the long term performance of VCD seems to exceed the baselines empirically.
>
> - **Lack of quantitative evaluation of causal discovery**: We thank the reviewer for the helpful suggestions and have included a quantitative analysis of disentanglement in the appendix. The results show that, compared to the RSSM representation, the VCD representation achieves better disentanglement, in accordance with our claim.
>
> - **Potential area(s) of improvement**: As discussed above, the main goal of our work is to address the problem of domain adaptation. We therefore considered it more appropriate to focus on the merits of an explicitly modular world model in the few-shot adaptation regime, as opposed to showing that VCD can predict more accurately than RSSM in the large data limit. For our purposes, we deliberately set up the experiments such that the baselines and VCD share the same encoder, decoder and transition functions in order to directly understand the effect of adding sparsity regularisation as fairly as possible. In terms of improving prediction accuracy, we believe that the proposed framework can be easily ‘upgraded’ by substituting the prediction architecture to something more sophisticated. As such, VCD will likely benefit from advances in the field of latent world models. In terms of generalising to richer environments, as we noted in the limitations section, we believe that one of the principal challenges for VCD is to relax the assumption that the causal graph is fixed over all time steps, as causal mechanisms (such as collisions) can be temporally local. This opens up an exciting line of research that can enable the use of causal world models in more complex and realistic environments.

---

### Official Review · Reviewer_qfCJ · 2022-10-24

**Confidence:** 4
**Correctness:** 3
**Technical Novelty And Significance:** 2
**Empirical Novelty And Significance:** 2
**Recommendation:** 3

**Clarity, Quality, Novelty And Reproducibility:**

**Clarity**: The paper presentation is very good, both in terms of writing and figures. My main gripe is that the lack of theoretical discussion makes it a bit harder to get a clear picture of the method.

**Quality**: The setting is very interesting and the presented VAE setup is solid. However, the method should really be demonstrated either in theory or practice. In its current form, the paper lacks a theoretical analysis and the empirical demonstration is not entirely convincing yet (see above).

**Novelty**: There is definitely novelty in this work, though I am still somewhat unsure how much exactly. There are not a lot of works on causal representation learning from interventional data and certainly not on the demonstration of more efficient adaptation to new settings. At the same time, the lack of theory and limited empirical analysis limits the novelty of this paper. Finally, it would be great if the authors could clarify the relation of their work to Ref. [1] a bit more.

**Reproducibility**: The work appears very reproducible – the authors have attached the code as supplementary material and promise to make it available publicly. The code quality (and level of documentation) looks great. There are also sufficient details in the appendix.

**References**:
- [1] Lachapelle et al, "Disentanglement via Mechanism Sparsity Regularization: A New Principle for Nonlinear ICA", CLeaR 2022, 2107.10098

**Strength And Weaknesses:**

**Strengths**:
1. The authors tackle an interesting and largely unsolved problem. Both learning latent causal structure from a different representation and demonstrating the benefits of that for rapid adaptation may be very impactful.
2. The model setup (a VAE with latent causal structure) makes sense.
3. It is great that the authors consider a weak form of supervision: the model is trained only on interventional data without any additional labels. This seems much more easily attainable than labelled data [e.g. 2] or counterfactual data [e.g. 3]
4. It is also great that the authors do not stop at the learning of causal structure, but demonstrate the benefits of that causal structure for more efficient finetuning on new settings.
5. The paper is clearly written and a pleasure to read.
6. The figure design is also good.
6. The authors have provided clean and well documented code along with the paper.

**Weaknesses**:
1. The setup makes some strong model assumptions, in particular that the causal graph is invariant over time steps and that there are no instantaneous causal effects. While the authors point this out, it would be good to stress these assumptions in the introduction and conclusions a bit more and to discuss them: why are they needed? Are they realistic? How could we relax them?
2. Relatedly, I have a hard time thinking of example systems that are compatible with all the assumptions here. It would be great if the authors could provide some motivating examples.
3. Probably my biggest criticism of the paper is that it is not clear what exactly the method should be able to achieve – there is essentially no theoretical discussion. In particular, the authors do not make any claims about identifiability. Up to which equivalence class should the causal variables, the causal graph, and the rendering function be identified? What is the role of the assumptions on the data-generating process for identifiability? What properties do the rendering function and the distributions need to satisfy? Footnote 1 points the reader to Ref. [1], but if the identifiability result of that paper applies here, that should really be discussed more explicitly (which is also important to judge the novelty of this paper).
4. The empirical evaluation also leaves something to be desired. It is okay that there is just a single toy dataset, but the performance in terms of causal representation learning should really be discussed quantitatively. There are many adequate metrics of disentanglement and graph similarity that could be applied here. The numbers that the authors do provide are not particularly impressive. In the appendix, the authors claim that it is "not trivial to obtain a way to map the [ground-truth factors] to a latent dimension". I would politely encourage the authors to have a look at how other representation learning papers deal with that – for instance, they could use a feature importance matrix like in Figure 7 to determine a map between ground-truth factors and learned latents and then compute graph edit distances.
5. The authors cite many relevant works, but missed Refs. [3-5] given below. In particular, it seems that Ref. [5] considers a very similar problem. This is clearly concurrent, so don't take this as a criticism of novelty, but it would make the paper stronger if you could mention and contrast their and your findings.

**References**:
- [1] Lachapelle et al, "Disentanglement via Mechanism Sparsity Regularization: A New Principle for Nonlinear ICA", CLeaR 2022, 2107.10098
- [2] Lippe et al, "CITRIS: Causal Identifiability from Temporal Intervened Sequences", ICML 2022, 2202.03169
- [3] Brehmer et al, "Weakly supervised causal representation learning", 2203.16437
- [4] Lippe et al, "iCITRIS: Causal Representation Learning for Instantaneous Temporal Effects", 2206.06169
- [5] Ahuja et al, "Interventional Causal Representation Learning", 2209.11924

**Summary Of The Paper:**

The authors propose a method for causal representation learning from interventional data and demonstrate how such a model can be adapted to new settings. They consider time-series data generated by a causal graphical model and a rendering function (e.g. to pixels); key assumptions include that there are no instantaneous causal effects and the causal structure is invariant over time steps. They then assume access to data from the observational and multiple interventional distributions. In this setting, a VAE with latent causal structure is trained. The authors argue that such a model should be able to adapt efficiently to data from a new environment provided the sparse mechanism shift hypothesis holds. This is then demonstrated empirically on toy data.

**Summary Of The Review:**

This work has a lot of potential: learning latent causal structure from interventional data on the pixel level could be very useful. The VAE setup makes sense. It is commendable that the authors do not stop at the identification of causal structure, but show that learning it is actually useful for predicting the future of a roll-out and for adapting to new settings.

Unfortunately, in its current form the paper does not demonstrate that the approach works. There should be either a theoretical discussion (like an identifiability result) or a convincing empirical demonstration that the model actually learns the correct causal structure. Right now, the paper has neither, and I can't recommend it for acceptance yet.

I encourage the authors to expand either of these two angles (or, ideally, both). If they can add such a demonstration, this could become a great paper and an impactful contribution to the field.

**Update after rebuttal**: Thanks to the authors for clear answers to my question and for addressing some of my criticisms in the upadted paper version. While the paper is improved, I still believe that it does not sufficiently demonstrate that the approach works. My score remains unchanged.

---

> ### Author Response · Authors · 2022-11-17
> **Response to reviewer qfCJ**
>
> We thank reviewer qfCJ for their thorough review and for acknowledging the potential of our work. We hope our response addresses the reviewer’s concern regarding the lack of theoretical discussion and empirical evaluations. Below we address the specific issues raised.
> - **Strong model assumptions**:
>   - *No instantaneous causal edges*: We believe that this assumption is realistic as most dynamical systems, when viewed in sufficient temporal resolution, should contain no instantaneous causal relationships. However, we also note that the assumption can be relaxed easily as DCDI does not require temporal knowledge and as such the model can handle instantaneous edges if required by the environment.
>   - *Invariant causal graph*: This assumption is necessary for our approach as the model learns one graph from all temporal transitions. We acknowledge that realistically this assumption might be violated by temporally sparse interactions such as collisions, where the causal graph might look different locally. We believe that this assumption could be relaxed, for example, by detecting local causal effects (e.g. [2]), though such investigations are beyond the scope of the present paper. We have identified this assumption as a key limitation of our current approach - this has been discussed in the limitations section of our manuscript.  Nevertheless, we believe that our work serves as a concrete and important first step towards the use of causal representations in world modelling and that the proposed framework of learning causal structures in the latent space jointly with world models would play a crucial part in adaptable systems that can deal with more complex environments.
> - **Lack of theoretical discussion**: We appreciate the reviewer’s comments and suggestions with respect to the theoretical basis of our work, and have taken these into consideration in the updated version of our paper. Here we hope to clarify further the relationship between VCD and [1]:
>   - [1] investigates the identifiability of a latent state-space model similar to ours. As noted in the updated ‘related works’ section of our paper, our work builds on the inductive biases developed in [1] and empirically explores the merits of such structured latent space models in terms of adaptation to new environments. VCD differs from the model studied in [1] in that it has a recurrent transition module which we found to be necessary for capturing long-term behaviours. As such, the identifiability results in [1] does not strictly apply to VCD, though they do offer strong intuitions as to why this class of method works in practice. To the best of our knowledge, VCD is the first to apply this class of latent SSM to pixel observations and, as noted by the reviewer, to show the adaptability of structured transition models.
>   - We therefore consider the main thrust of our work to be empirical in nature, although we also acknowledge the importance of identifiability results in claiming causality. In our view, the main goal of our work is to leverage the sparsity of mechanism changes to transfer knowledge between environments, rather than the recovery of ground truth variables - to this end, experiments reported in the present paper demonstrate that having a structured transition model does indeed improve the model’s ability to predict in new environments significantly. Against this backdrop, we believe that our work serves as an important step towards bridging the theoretical framework of disentanglement via mechanism sparsity and practical use-cases such as sim-to-real transfer.
>
> - **Lack of quantitative evidence for disentanglement**: We thank the reviewer for their suggestion to improve our empirical evaluation and have included a quantitative analysis of the disentanglement along with our comments in Appendix D.2. We hope that this provides quantitative evidence that VCD can indeed learn disentangled representations and capture causal mechanisms.
> - **Missing references**: We appreciate these suggestions and have taken them into account in the updated related works section.
>
> [1] Lachapelle et al. (2022) “Disentanglement via Mechanism Sparsity Regularization: A New Principle for Nonlinear ICA”
>
> [2] Seitzer et al. (2021) “Causal influence detection for improving efficiency in reinforcement learning”

---

> > ### Comment · Reviewer_qfCJ · 2022-11-18
> > **Thanks**
> >
> > Thanks to the authors for their detailed response and the updated manuscript. I appreciate in particular the clarification about the the scope of the paper and the added metrics. These changes have improved the paper.
> >
> > Nevertheless, I still believe that the paper does not sufficiently demonstrate that the approach works. As the authors point out, the identifiability theory of Lachapelle et al does not strictly apply here. That would be fine if there was a strong empirical demonstration, but the experiments are limited to a few toy scenarios and the results are not entirely convincing, at least to me. Overall, I think that this direction of research is interesting and relevant, but still needs more work before acceptance.

---

> > > ### Author Response · Authors · 2022-11-18
> > > **Reply to reviewer qfCJ**
> > >
> > > We thank the reviewer for the prompt reply and acknowledging the improvements made in the revised manuscript.
> > >
> > > The main thrust of our work is empirical in nature. We firmly believe that our work serves to bridge theory and application of causality in ML. As such, we define success of our model both, in terms of the causal graph recovered as well as in the model’s ability to adapt. Specifically, we demonstrate that the quality of the causal graph discovered suffices to effect efficient adaptation to novel environments. This in itself presents a novel outlook and capability. As such no direct prior work exists. However, our model outperforms competitive baselines in terms of significantly improved few-shot adaptation on test environments. We acknowledge that the experiments are limited to toy scenarios. However, as also noted by reviewer Tfgu, the specific setting we choose is of comparable complexity to recent papers in the field. We would therefore argue that the complexity of our environments indeed corroborates the significance of our results. In our view, there has been extensive work on causal representation learning and causal discovery, but to the best of our knowledge, VCD is the first practical method to actually leverage causal representations and models to perform domain adaptation. Nevertheless, causal representation learning is a rapidly expanding field with theoretical and empirical advances that improve the quality of disentanglement. We believe that the framework proposed here will be instrumental for systems that can tackle more complex environments as the quality of disentanglement continues to improve.
> > >
> > > We appreciate the reviewer’s view that our work is interesting and relevant and believe that in the current version we have addressed all the actionable concerns the reviewer has raised. It is regrettable, that the reviewer nevertheless remains unconvinced. However, it is not clear what “more work before acceptance” would actually entail. Are there concrete improvements you would suggest beyond your original review?

---

### Official Review · Reviewer_xUbv · 2022-10-24

**Confidence:** 4
**Correctness:** 4
**Technical Novelty And Significance:** 3
**Empirical Novelty And Significance:** 2
**Recommendation:** 6

**Clarity, Quality, Novelty And Reproducibility:**

Clear, most of the details are deferred to appendices. Novelty is in the use of latent variables and in the analysis of this particular experiment

**Strength And Weaknesses:**

Strengths:
- The addressed problem is highly relevant. The above-mentioned list of properties seems necessary for artificial intelligence and few works check all these boxes.
- The experiments show the advantages of the current proposal and even show the emergent property of axis alignment.

Weaknesses:
- I'd like to see more justification as to why this model should be called "causal", as opposed to simply "predictive and modular, while encouraging sparsity". While it seems reasonable that a predictive, modular, sparse model would capture causal dependencies, is there a formal reason why it couldn't capture consistent temporal correlations? What would happen if I were to provide as evidence a video in which entropy increases, but run backwards (so that entropy decreases)? What if I were to show any other video with consistent correlations but lack of causation?
- The experiments are very limited (just one toy example), and the proposed model is not studied in detail:
  - In particular, the dynamics seem pretty deterministic. How does it handle uncertain actions? Can it handle actions in which the target state is multimodal given a source state?
  - How would this model handle situations in which the uncertainty gives rise to multimodality? For instance, imagine that there's a very small wall in the middle of a room and a ball can bounce off it only in a concrete trajectory, but won't if the trajectory deviates slightly. In this case, when predicting in open loop (observations are not fed back into the model), will it be able to handle the multimodality of the predictions properly?
- Other
  - Format violation: More than 9 pages
  - I believe |G| is never defined

**Summary Of The Paper:**

In this work a model for sequences with the following properties is proposed:
- Modular description of the transition function
- Handling of interventions, defined as the modification of a target of the transition function
- Learning of the dependency structure
- Use of latent variables

The authors build on top of the formulation from "Differentiable Causal Discovery with Interventional data" (DCDI), which exhibits the above properties, except for the latent variables. The addition of the latent variables introduces intractability that is solved through the use of a variational inference.


**Summary Of The Review:**

A step in a good direction, drawing most of the insights from existing work, limited empirical evaluation.

---

> ### Author Response · Authors · 2022-11-17
> **Response to reviewer xUbv**
>
> We thank reviewer xUbv for their time and the thorough review. We are delighted that they share our position regarding the relevance of the addressed problem. Below we address the issues raised.
> - **Why “causal”**: We use the term “causal” in accordance with the causal discovery literature, wherein the conditions and assumptions under which true causal relationships can be discovered have been extensively discussed, see e.g. [1]. Importantly, our model explicitly utilises the notion of interventions (and the invariance of causal mechanisms under them), which we believe is only afforded by a causal interpretation of the model. Whilst we do not claim the theoretical identifiability of causal mechanisms in our application, the structure discovery module is firmly grounded in the causal discovery literature, by using the inductive bias of sparsity. We therefore decided it would be appropriate to name the proposed model “causal” to reflect the main motivation behind our work.
>
>
> - **Limited experimental evidence**: For the purpose of demonstrating the adaptability of causal models, the experiment environments are chosen such that the ground truth causal relationships are unambiguous. We acknowledge that in its current form, the model might not be able to cope with complex environments with challenges such as multimodality or sparse interactions (as in the case of a small wall in the middle). However, we note that these limitations are inherited from the prediction module (i.e. RSSM) due to the assumption of gaussian distributions. We emphasise that the proposed framework can be readily modified to suit more complex environments, by substituting the prediction module with any other architecture that is compatible with ELBO training. We agree it would be interesting to investigate how complex models would interact with the proposed framework and have added a related discussion in the limitations section.
>
> [1] Brouillard, S. et al. (2020) “Differentiable causal discovery from interventional data”

---

### Official Review · Reviewer_gksh · 2022-10-26

**Confidence:** 4
**Correctness:** 3
**Technical Novelty And Significance:** 2
**Empirical Novelty And Significance:** 2
**Recommendation:** 3

**Clarity, Quality, Novelty And Reproducibility:**

### Clarity
The paper is very easy to read.

### Quality
The proposed method seems sensible in theory

### Reproducibility
The provided code appears to make reproducing the results easy.

### Novelty
The method is closely related to prior work. In particular, [1] addresses a very similar problem. The only difference it that the latent causal model is learned via DCDI, so that the different environments don't need different labels on the interventions.

**Strength And Weaknesses:**

Strength:
- The paper is slightly more applicable than [1], as it doesn't require each environment to have labelled intervention targets.
- The paper is clearly written.
- The authors have provided code with their paper.

Weakness:
- The empirical evaluation is only done on a toy dataset of coloured circles. This is insufficient to assess the performance of the proposed model.
- The paper has no identifiability guarantees.
- The paper is very similar to [1], but this is not fairly reflected in the text.
- In table 2, it appears like the graph and intervention targets aren't learned very accurately.

[1] Lippe, Phillip, et al. “CITRIS: Causal Identifiability from Temporal Intervened Sequences.” 2022

**Summary Of The Paper:**

The authors propose a method for latent dynamics modelling. In latent space, the latent variables are assumed to affect variables in the next time step via a sparse graph interaction. The model allows for soft interventions on the mechanisms. The model is trained via the ELBO and the graph structure, causal mechanisms and interventino targets are trained via the method from Brouillard et al. (2020). The method is evaluated on a toy image dataset of moving balls.

**Summary Of The Review:**

The paper in limited in technical novelty and has neither theoretical guarantees, nor has been convincingly shown to work well in practice.

---

> ### Author Response · Authors · 2022-11-17
> **Response to reviewer gksh**
>
> We thank reviewer gksh for their time and helpful comments. We would like to draw particular attention to the criticism regarding the lack of novelty and the similarity of our work to [1]. As we suggest below, we believe that the two works address a fundamentally different problem setting, and we therefore politely encourage the reviewer to reconsider their assessment of the novelty of our work. Below we address the specific issues raised.
>
> - **Lack of identifiability results**: We acknowledge the importance of identifiability results in claiming causality, although we also note that the main thrust of our work is empirical in nature. The identifiability guarantees for a similar SSM to ours have already been extensively discussed in previous work [4]. As noted in the updated ‘related works’ section of our paper, our work builds on the inductive biases developed in [4] and empirically explores the merits of such structured latent space models in terms of adaptation to new environments. To the best of our knowledge, our work is the first model to have successfully employed this class of latent SSM to pixel-based environments and showed its adaptation capability. Moreover, the main goal of our work is to leverage the sparsity of mechanism changes to transfer knowledge between environments, rather than the recovery of ground truth variables - experiments reported in the present paper demonstrate that having a structured transition model does indeed improve the model’s ability to predict in new environments significantly. Importantly, our model significantly outperforms the baselines even when the learnt causal graphs and intervention targets are imperfect, which corroborates our claim regarding the merits of structured transition models. Against this backdrop, we believe that our work serves as an important step towards bridging the theoretical framework of disentanglement via mechanism sparsity and practical use-cases such as sim-to-real transfer. In any case, we appreciate the reviewer’s attention to identifiability guarantees of our work, and have updated our manuscript to situate the present work within the theoretical framework of [4] accordingly.
>
> - **Similarity to [1]**: We would like to highlight that while VCD appears to share a similar spirit with CITRIS [1] in learning latent structures using a VAE, the two works address a fundamentally different problem, and as such, we believe that the parallels between the two works are limited.
> Problem setting: A critical difference between VCD and CITRIS lies in the formulation of interventions. VCD considers interventions as changes in variable dynamics between different environments, e.g. removal of spring forces from one environment to another. CITRIS, on the other hand, considers interventions as direct, local changes to some latent variables at each time step, e.g. rotation of an object by an agent within a given temporal sequence. This difference is crucial as it means that the two works have diverging implications: by considering dynamical shifts across environments, VCD tackles the problem of domain adaptation, whereas CITRIS addresses whether and how causal variables may be identified. By extension, the main objective of our work is to provide empirical evidence demonstrating that VCD can transfer knowledge across environments (as suggested by reviewer Tfgu), whereas that of CITRIS is theoretically-inclined, i.e. to show that identifiable causal representations can be learned. We believe that the ability to utilise causal knowledge to adapt to new settings has not been investigated by other works - as pointed out by reviewer qfCj and to the best of our knowledge - and we argue that such capabilities are essential to flexible and adaptable systems, e.g. sim-to-real transfer, where shifts in underlying scene dynamics are common. Nevertheless, we appreciate the reviewer’s comment that this is not sufficiently reflected in the text, and have updated our manuscript accordingly.
>
> - Another difference between VCD and CITRIS concerns whether interventions are labelled, as the reviewer suggested. Unlike CITRIS, VCD does not require labelled interventions and therefore relies on a weaker form of supervision.

---

> > ### Comment · Reviewer_gksh · 2022-11-24
> > **Agreed with dissimilarity to [1], still unconvinced**
> >
> > I thank the authors for their reply. I agree that the interventional regime in CITRIS is temporal, while VCD has multiple datasets with different intervention settings. Still, the paper does not have theoretical novelty and for a purely empirical paper, I find its evaluation on a toy dataset with coloured balls too limited. My score remains unchanged.

---

> ### Author Response · Authors · 2022-11-17
> **Response to reviewer gksh - 2**
>
> - **Limited empirical evaluation and toy dataset**: We note that our evaluation environment is commensurate in complexity with related works such as [2, 3]. The rationale for using such environments is that they are designed with a ground-truth causal structure.  An environment with a clear ground-truth causal structure is particularly important in the context of our work as it enables valid comparison against learnt causal graphs from our model. Indeed, experiment results reported in the appendix of the present paper show, by comparing ground-truth and learnt causal graphs, that our model is able to learn the correct causal structure. Hence, the design of our environment was motivated by two key principles: i) valid and effective evaluation between ground-truth and learnt structures, and ii) incorporation of sufficiently varied mechanisms and intervention targets whilst ensuring i). We believe that this is why environments with similar levels of complexity are commonly used in related works [e.g. 2, 3]), as has also been suggested by reviewer Tfgu. In view of the above, we believe the new capabilities developed in our work, i.e. learning disentangled and discrete mechanisms, are unique to our approach and will play a crucial role in systems that can efficiently adapt in complex environments.
>
> - **Inaccurate graph and interventions**: Whilst the learnt intervention targets and causal graph are indeed imperfect, the main takeaway from our experiments is that the model’s adaptation ability significantly outperforms the baselines despite the extra edges in the graphs, as discussed above in the identifiability discussion. Moreover, upon close inspection of the learnt causal graph, we observe that the missed edges correspond to non-linear terms in the ground-truth dynamics of the system which only had small effects on the trajectories of the particles. This is discussed in more detail in the paragraph following table 3 in the appendix.
>
> [1] Lippe et al. (2022) “CITRIS: Causal Identifiability from Temporal Intervened Sequences.”
>
> [2] Goyal et al. (2020) “Object Files and Schemata: Factorizing Declarative and Procedural Knowledge in Dynamical Systems”
>
> [3] Assouel et al. (2022) “VIM: Variational Independent Modules for Video Prediction”
>
> [4] Lachapelle et al. (2022) “Disentanglement via Mechanism Sparsity Regularization: A New Principle for Nonlinear ICA”

---

### Official Review · Reviewer_LfHe · 2022-10-31

**Confidence:** 3
**Correctness:** 3
**Technical Novelty And Significance:** 3
**Empirical Novelty And Significance:** 3
**Recommendation:** 5

**Clarity, Quality, Novelty And Reproducibility:**

Overall the presentation has great clarity and the figures of the experiment results are of good quality. The details of the model are described and the code is provided in the supplemental materials.

**Strength And Weaknesses:**

Strengths:

1. Modularity: When encountered with new and unseen environments, VCD can quickly adapt by training the VCD on trajectories in the new environment while fixing the trained parameters for the causal graph and the mechanisms in the undisturbed environment.

2. Sample efficiency: During adaptation, VCD is more sample efficient as compared to the baselines RSSM and MultiRSSM.

3. Extensive experiments are done in the multi-body environment.

Weakness:

1. Typos in equation 14: p(z^t|o^t)  should be q(z^t|o^t)

2. The experiment is only tested on one simple environment.  Lack of complex environment to demonstrate its claimed effect on higher-dimensional raw image input.

3. The causal graph is learned in the latent space and depends on the dimension d of the latent space. It would be interesting to provide comments on the choice of d, and the relationship between d and the actual number of edges between entities in the multi-body system.

4. The multi-entity environment only has 4 entities. How will the model scale with a higher number of interacting objects in the task and a higher number intervene objects?



**Summary Of The Paper:**

This paper presents VCD, a world model that can modularly transfer knowledge between different interventional environments. It causally factories the transition model in the latent space and jointly learns an approximate posterior model, transition model, causal graph, and the intervention targets together with a derived ELBO loss. VCD learns both observational and interventional mechanisms which enable modular adaptation between different interventional environments, as it taking advantage of the invariances of causal mechanism.

**Summary Of The Review:**

Given the above strengths and shortcomings of this paper, I recommend this paper be considered slightly below the acceptance threshold.

---

> ### Author Response · Authors · 2022-11-17
> **Response to reviewer LfHe**
>
> We thank reviewer LfHe for their time and thorough review. Below we respond to the issues raised.
>
> - **Typo**: We thank the reviewer for pointing this out. This has been rectified.
>
> - **Complexity of environment**: We note that our evaluation environment is commensurate in complexity with related works such as [1, 2]. The rationale for using such environments is that they are designed with a ground-truth causal structure.  An environment with a clear ground-truth causal structure is particularly important in the context of our work as it enables valid comparison against learnt causal graphs from our model. Indeed, experiment results reported in the appendix of the present paper show, by comparing ground-truth and learnt causal graphs, that our model is able to learn the correct causal structure. Hence, the design of our environment was motivated by two key principles: i) valid and effective evaluation between ground-truth and learnt structures, and ii) incorporation of sufficiently varied mechanisms and intervention targets whilst ensuring i). We believe that this is why environments with similar levels of complexity are commonly used in related works [1, 2]), as has also been suggested by reviewer Tfgu. In view of the above, we believe the new capabilities developed in our work, i.e. learning disentangled and discrete mechanisms, are unique to our approach and will play a crucial role in systems that can efficiently adapt in complex environments.
>
> - **Relationship between latent dimension and actual dimension**: Intuitively, if d is smaller than the actual dimensionality of the system, the model would degrade since the latent space is not big enough. On the other hand, if d is larger than the actual dimensionality, we expect that the sparsity regularisation would encourage some of the variables to be dummy variables that capture no information about the scene (and as such has no causal edges). Empirically, we observe this to be the case in our experiments. In the revised appendix, we show the correlation between the learnt latents and the ground-truth state variables and it can be seen that some variables do not correlate with any state variables at all (Fig. 7, 8), corroborating our above suggestion that these variables have been reduced to ‘dummy’ variables.
>
> - **Scalability to more objects**: In the model’s current form, since the number of possible edges in the causal graph scales with the square of the number of variables, scaling to a higher number of objects could be difficult. Our response to this limitation of our model has been discussed in the limitations section of our paper: we believe that the proposed framework can be naturally combined with object-centric generative models which could improve the scalability by, for example, considering pair-wise interactions. We believe that the proposed framework of discovering causal mechanisms in the latent space will be a crucial component of such systems.
>
> [1] Goyal et al. (2020) “Object Files and Schemata: Factorizing Declarative and Procedural Knowledge in Dynamical Systems”
>
> [2] Assouel et al. (2022) “VIM: Variational Independent Modules for Video Prediction”

---

### Decision · Program_Chairs · 2023-01-20

**Decision:**

Reject

**Justification For Why Not Higher Score:**

Experiments are too simple and limited theoretical novelty.

**Justification For Why Not Lower Score:**

NA

**Metareview: Summary, Strengths And Weaknesses:**

The paper presents a modular world model that can transfer knowledge between different interventual environments. It factorizes the latent space due to causality. The model is trained using VI. The method is evaluated on a toy dataset.

Strength and weaknesses. The idea of modular world models is nice and the model shows also better sample efficiency then standard RSSMs. Yet, as pointed out by reviewer gksh  the approach is very similar to other methods and hence does not offer a lot of theoretical contribution. Further, the experimental evaluation was considered to be not sufficient by almost all reviewers. Authors are encouraged to extend their evaluation to more complex environments and resubmit this interesting line of work to another conference.

**Summary Of Ac-Reviewer Meeting:**

N/A